



# Next-generation ice nucleating particle sampling on aircraft: Characterization of the High-volume flow aERosol particle filter sAmpler (HERA)

Sarah Grawe[1], Conrad Jentzsch[1,2], Jonas Schaefer[1], Heike Wex[1], and Frank Stratmann[1]

[1]Leibniz Institute for Tropospheric Research, Leipzig, Germany
[2]Now at: Leipziger Verkehrsbetriebe, Leipzig, Germany

**Correspondence:** Sarah Grawe (grawe@tropos.de)

**Abstract.**

Atmospheric ice nucleating particle (INP) concentration data from the free troposphere are sparse, but urgently needed to understand vertical transport processes of INPs and their influence on cloud formation and properties. Here, we introduce the new High-volume flow aERosol particle filter sAmpler (HERA) which was specially developed for installation on research aircraft and subsequent offline INP analysis. HERA is a modular system constisting of a sampling unit and a powerful pump unit and has several features which were integrated specifically for INP sampling. Firstly, the pump unit enables sampling at flow rates exceeding 100 L min$^{-1}$, which is well above typical flow rates of aircraft INP sampling systems described in the literature (~10 L min$^{-1}$). Consequently, required sampling times to capture rare, high-temperature INPs ($\geq$-15 °C) are reduced in comparison to other systems and potential source regions of INPs can be confined more precisely. Secondly, the sampling unit is designed as a seven-way valve, enabling switching between six filter holders and a bypass with one filter being sampled at a time. In contrast to other aircraft INP sampling systems, the valve position is controlled remotely via software so that manual filter changes in-flight are eliminated and the potential for sample contamination is decreased. This design is compatible with a high degree of automation, i.e., triggering filter changes depending on parameters like flight altitude, geographical location, temperature, or time. In addition to the design and principle of operation of HERA, this paper presents laboratory characterization experiments with size-selected test substances, i.e., SNOMAX® and Arizona Test Dust. The particles were sampled on filters with HERA, varying either particle diameter (300 nm to 800 nm) or flow rate (10 L min$^{-1}$ to 100 L min$^{-1}$) between experiments. The subsequent offline INP analysis showed good agreement with literature data and comparable sampling efficiencies for all investigated particle sizes and flow rates. Furthermore, the deposition efficiency of atmospheric INPs in HERA was compared to a straightforward filter sampler and good agreement was found. Finally, results from the first campaign of HERA on the High Altitude and LOng range research aircraft (HALO) demonstrate the functionality of the new system in the context of aircraft application.



## 1 Introduction

Ice nucleating particles (INPs) have been a focus of atmospheric science for several decades due to their effect on primary ice formation in clouds. While pure cloud droplets freeze homogeneously at ~-37 °C (Pruppacher and Klett, 1997), the freezing onset is shifted towards higher temperatures in the presence of INPs. With that, INPs influence cloud properties such as the radiative effect and lifetime, as well as precipitation formation (Creamean et al., 2013; Michaud et al., 2014; Vergara-Temprado et al., 2018; Lin et al., 2022). An accurate representation of INP concentrations, i.e., the number of INPs active at a certain temperature per volume of air, could help decrease the currently large uncertainty of the effect of clouds and aerosol-cloud-interactions on Earth's radiative budget in climate models (Forster et al., 2021). Using aerosol particle properties to predict INP concentrations is subject to ongoing research (Phillips et al., 2013; DeMott et al., 2015; Fitzner et al., 2020), albeit a difficult task, since it is still not completely understood what makes certain particles more efficient at nucleating ice than others. In any case, a sound database for the verification of parametrizations of INP concentrations is necessary which requires atmospheric measurements of INP concentrations. Especially remote locations such as the Arctic, Antarctica, the Southern Ocean, and the free troposphere have not yet been sufficiently studied to provide conclusive INP parametrizations (Murray et al., 2021).

Nonetheless, the amount of INP concentration data generated has increased tremendously in recent years first and foremost due to the development of a large number of different instruments for offline immersion freezing characterization (DeMott et al., 2011, 2018). In contrast to complex online instrumentation, e.g., continuous flow diffusion chambers (CFDCs; Rogers, 1988; Stetzer et al., 2008; Garimella, 2016), these are relatively easy to setup and use several orders of magnitude larger sampling volumes, enabling the investigation of rare, high-temperature INPs (≥-15 °C) which are not captured by online instruments. Some offline techniques operate with microliter-sized droplets on glass substrates (Budke and Koop, 2015; Whale et al., 2015; Chen et al., 2018) or in separate wells (Conen et al., 2012; Hill et al., 2014) and usually cannot produce meaningful data below ~-30 °C due to freezing induced by impurities (measurement background). Others use nano- or picoliter-sized droplets which shifts the freezing onset temperature of pure water droplets towards the homogeneous freezing limit (Pummer et al., 2012; Wright and Petters, 2013; Peckhaus et al., 2016; Stan et al., 2009; Riechers et al., 2013; Reicher et al., 2018). All of these techniques can be operated with aqueous suspensions such as collected sea (Wilson et al., 2015; Irish et al., 2017), river (Knackstedt et al., 2018; Moffett, 2016), or cloud water (Joly et al., 2014), precipitation samples (Petters and Wright, 2015), impinger samples (Šantl-Temkiv et al., 2017), impactor samples (Mason et al., 2016), or washing water of filter samples (McCluskey et al., 2018; Adams et al., 2020; Hartmann et al., 2021; Jakobsson et al., 2022). Some offline instruments also use punched-out pieces of filter material with collected aerosol particles immersed in water (Conen et al., 2012; Welti et al., 2018). By using a combination of offline instruments featuring different droplet sizes, it is possible to span a broad range of INP concentrations in a temperature regime of which only the lowermost bound can be covered with the online techniques.

Automatic aerosol particle sampling equipment is commercially available, low-maintenance, and hence operated frequently in ground- or ship-based measurement campaigns and in long-term measurements to obtain INP concentrations (Schrod et al., 2020; Schneider et al., 2021; Testa et al., 2021; Sze et al., 2022). While ground-based aerosol particle sampling is an important step towards revealing the nature and sources of INPs, open questions exist concerning the mechanisms making INPs airborne,



the vertical transport of INPs, their concentrations at cloud level, and their influence on cloud formation and properties (Coluzza et al., 2017). Furthermore, the influence of cloud processing on INP concentrations and the relative abundance of INPs in cloud particle residuals have rarely been investigated (Stopelli et al., 2015; Levin et al., 2019). In-situ measurements of free tropospheric INPs are generally sparse, as they can only be performed on mountain sites (DeMott et al., 2003a; Lacher et al., 2018; Conen et al., 2022) or with the help of airborne platforms. Creamean et al. (2018) and Porter et al. (2020) describe aerosol particle sampling for ice nucleation analysis with the help of tethered balloons, which, in contrast to a stationary measurement site, offer flexibility regarding the sampling altitude but have restricted payloads of a few kilograms at most. The same holds for aerosol particle samplers deployed on small unmanned aerial vehicles (Schrod et al., 2017; Jimenez-Sanchez et al., 2018; Bieber et al., 2020). An alternative to the described approaches are INP measurements on research aircraft which can reach the upper troposphere and are typically equipped with a large instrument suite for answering specific research questions. Consequently, there are simultaneous measurements of, e.g., meteorological parameters, aerosol particle properties, and trace gases, which can contribute to the interpretation of the INP results. Both online methods, i.e., CFDCs (Rogers et al., 1998, 2001; DeMott et al., 2003b; Levin et al., 2019; Barry et al., 2021b), and offline methods, i.e., filter sampling systems (Bigg, 1967; Flyger et al., 1973; Borys, 1989; DeMott et al., 2016; Price et al., 2018; Levin et al., 2019; Sanchez-Marroquin et al., 2020, 2021; Varble et al., 2021; Barry et al., 2021b), have been used on aircraft. Online methods provide the benefits of better time resolution compared to filter samples and the possibility to investigate different nucleation modes depending on the thermodynamic conditions in the measurement chamber. Unfortunately, changing conditions takes some time, with the duration depending on the planned temperature and/or humidity step, which restricts flexibility (Rogers et al., 2001). Furthermore, most online instruments work with low flow rates of ~1 L min$^{-1}$ (Rogers, 1988; Stetzer et al., 2008; Garimella, 2016), i.e., high time-resolution data are restricted to below ~-25 °C where INP concentrations are above the detection limit. An additional disadvantage are the large dimensions of online instrumentation which can conflict with common space- and weight restrictions on aircraft. In contrast, aerosol particle filter samples can be collected with comparably small, light-weight equipment. As they are suitable for generating INP concentration data between 0 and ~-30 °C, and even down to -37 °C when nanoliter-sized droplets are used, they are a valuable addition to online INP measurements on aircraft.

All of the above mentioned studies describing aerosol particle filter sampling on aircraft for offline INP analysis use commercially available filter holders or modifications of those, which are exposed to ambient air from the outside of the aircraft via an inlet system and sampling line. Changing filters in-flight involves manual valve operation, removal of the sampled filters within their holders, and insertion of previously prepared filter holders with clean filters (Bigg, 1967; Flyger et al., 1973; Borys, 1989; DeMott et al., 2016; Price et al., 2018; Levin et al., 2019; Sanchez-Marroquin et al., 2020, 2021; Varble et al., 2021; Barry et al., 2021b). This approach comes with several drawbacks. Firstly, there is no possibility of automation and an on-board operator has to perform the filter changes. Secondly, contamination could be introduced to the samples during the handling of the filter holders in-flight. Last but not least, removing equipment from the aircraft in-flight is not always allowed from an aviation certification point of view. The collection of field blanks on aircraft, which are essential for estimating background levels in the immersion freezing experiments, has been described by Borys (1989), Levin et al. (2019), Barry et al. (2021b), and Sanchez-Marroquin et al. (2021). The blanks were handled in the same way as the filter samples, i.e., prepared in the laboratory, placed



inside a clean filter holder, and connected to the sampling line in-flight but without air exposure. However, in past campaigns the blanks were not taken during every flight, i.e., not every sampled filter had a corresponding blank. Concerning volumetric flow rates through the filters, values of ~10 L min$^{-1}$ (Borys, 1989; DeMott et al., 2016; Sanchez-Marroquin et al., 2019) or less (Levin et al., 2019; Barry et al., 2021b) are reported. One exception is the study by Flyger et al. (1973), who sampled at a rate of more than 50 L min$^{-1}$. Generally, a high flow rate is desirable as INP numbers above the measurement background can be collected in a shorter period of time. Consequently, more filters can be sampled per flight and there is an increase in temporal and spatial resolution. In previous studies, flow through the filters was generated by pumps downstream of the filter holders supported by the ram pressure of the moving aircraft (Flyger et al., 1973; Price et al., 2018; Sanchez-Marroquin et al., 2020, 2021). However, none of the setups included active control of the pump speed which would be another step towards automation and would make the systems more versatile for isokinetic sampling on a range of different aircraft with differing inlet and sampling line designs.

In this paper, we describe the design and performance of the novel High-volume flow aERosol particle filter sAmpler (HERA) which was specially developed for aircraft application and offline INP analysis. In contrast to the above mentioned sampling methods, HERA is highly automated. Up to six filters can be loaded into the device prior to takeoff and selected in-flight via an electric motor controlled by software. This design eliminates manual filter handling and lowers the potential for contamination. One of the six slots can be reserved for a field blank for background correction. HERA also features a powerful, actively-controlled pump unit downstream of the filters which can generate flow rates exceeding 100 L min$^{-1}$, depending on the selected filter medium and the pressure conditions. A prototype of HERA was successfully deployed during PAMARCMiP (Polar Airborne Measurements and Arctic Regional Climate Model Simulation Project) in late winter 2018 (Hartmann et al., 2020). Afterwards, the system was revised and characterized in the laboratory and field. In the following, we present the technical description of HERA, characterization experiments with standard and atmospheric INPs, and first results from sampling of HERA on aircraft during the HALO (High Altitude and LOng range research aircraft) mission CIRRUS-HL (cirrus in high latitudes). Materials and methods are described separately in the upcoming three chapters, followed by the results and their discussion.

## 2 Instrument description

### 2.1 Design

HERA was conceptualized and built by enviscope GmbH (Frankfurt, Germany) in close collaboration with TROPOS. Figure 1 a shows a schematic of the installation on aircraft using the example of the HALO CIRRUS-HL mission. HERA consists of a sampling unit and pump unit. The sampling unit is connected to an inlet, in case of HALO the HALO Submicrometer Aerosol Inlet (HASI), through which ambient aerosol particles are collected. If available, as during CIRRUS-HL, HERA can also sample from a second inlet, e.g., a Counterflow Virtual Impactor (CVI; Ogren et al., 1985; Mertes et al., 2007) for in-cloud sampling of residual particles. In this case, electrical valves are installed and controlled via software to open/close the connection to the respective inlets.



The sampling unit houses an inset containing six metal filter holders which, together with a bypass tube, are arranged
concentrically around a shaft connecting two seven-way valves (see photo and cross section in Fig. 1 b). The valves are turned
in unison via a chain drive connected to a servo motor and the valve position is set remotely via software. As a result, air flows
through two 90° bends onto one distinct filter. The valve construction involved careful consideration of design and materials
to avoid leaks among the filter positions and from HERA to the ambient environment (see Appendix A for more details).
Two sets of temperature and pressure sensors prior and post filter record the thermodynamic conditions in-line. Furthermore,
the sampling unit contains the data acquisition and control computer. The pump unit is equipped with three oil-free vacuum
scroll pumps (SVF-E0-50PF, Scroll Labs, USA). Each of them is able to generate a flow rate of 50 L min$^{-1}$ for undisturbed
standard conditions, i.e., without any filter medium upstream of the pumps. Generally, HERA can be operated with different
filter media, such as quartz fiber filters or polycarbonate (PC) membrane filters, with a diameter of 47 mm. In this work, we
only present experiments with PC filters (Nuclepore$^{TM}$ Track-Etched Membranes, Whatman, UK) which are frequently used
for INP sampling due to their smooth surface from which particles can be washed off with high efficiency (e.g., DeMott et al.,
2016; Tarn et al., 2018). Furthermore, PC filters are chemically inert, making them suitable for pre-treatments (Hill et al.,
2017). At a constant inlet flow rate and unchanging number of pores, a smaller pore size filter always leads to a larger pressure
drop and hence an increase in pump speed in comparison to a larger pore size filter (Liu and Lee, 1976; Zíková et al., 2015).
This relation, together with the inlet-specific flow rate requirements must be kept in mind when selecting the filter medium. See
Sec. 3.1 for a detailed discussion of the effect of PC filter pore size on INP sampling. The pump unit also contains a mass flow
meter (4043, TSI Incorporated, USA) upstream of the pumps whose data, together with the in-line temperature and pressure
measurements in the sampling unit, is used to calculate the volumetric flow rate at the HERA inlet. The maximum error in flow
rate, as estimated by error propagation utilizing the manufacturer-specific accuracies of the temperature and pressure sensors
and the flow meter, is ~3 %. The pumps are actively controlled to keep the inlet volumetric flow rate constant independent of
145 pressure and temperature changes. The flow rate is controlled remotely via software to maintain a setup-specific value (e.g.,
40 L min$^{-1}$ at the HASI during CIRRUS-HL) and set to zero during turning the valve to select a new filter position.

## 2.2 Theoretical sampling characteristics

The general goal when sampling aerosol particles is the minimization of particle losses and enrichment, so that the collected
particles are comparable to the ambient aerosol in terms of their physicochemical properties. The overall sampling efficiency
is influenced by the aspiration efficiency of particles in the inlet and the transmission efficiency in the tubing, both of which are
strongly dependent on the particle size and mass (Brockmann, 2011). Generally, small particles are prone to diffusional losses,
whereas large particles are lost due to inertial and gravitational forces. INP sampling specifically calls for the representative
sampling of particles in the size range above 0.5 $\mu$m, whose occurrence has been shown to correlate with measured INP
concentrations (DeMott et al., 2010; Testa et al., 2021). Care must be taken to sample isokinetically, i.e., to align the inlet in the
155 main wind direction and to adapt the sample flow velocity to the wind speed. Especially the latter is a challenge on aircraft, as
the velocity of the aircraft relative to the air mass, i.e., the true air speed (TAS), usually varies with flight altitude. If the sample
flow velocity is lower than the TAS, sampling is sub-isokinetic and particles with a sufficiently large inertia are over-sampled.



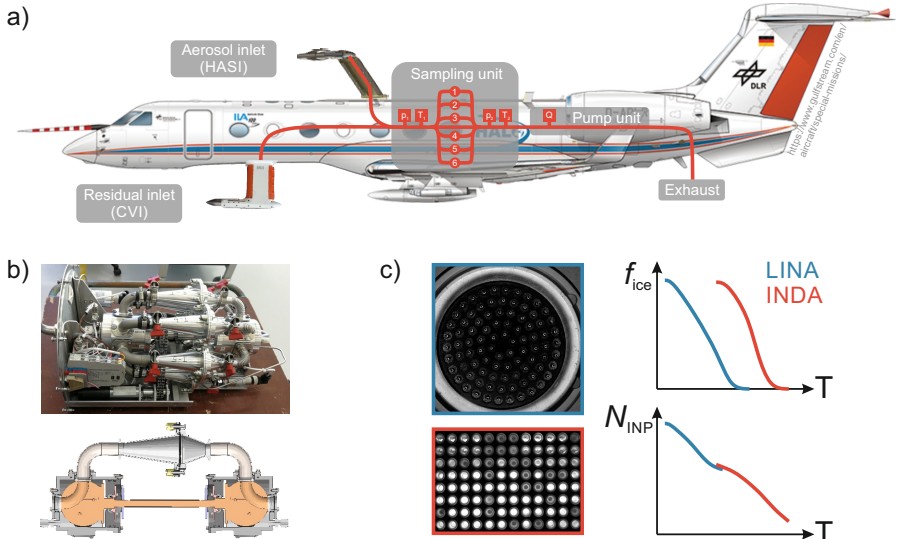

**Figure 1.** a) Installation of HERA on the HALO research aircraft. The sampling unit can be connected to either the aerosol particle inlet (HASI) or the residual particle inlet (CVI) for sampling outside or inside clouds. Switching between inlets is performed remotely with electrical valves. b) Top: Photo of the filter holder inset of the HERA sampling unit. Bottom: Cross section of one of the six filter holders connected to the seven-way valves. c) Cold stage setups for the immersion freezing analysis of the HERA filter extracts: LINA (framed in blue, 1 µL droplets), INDA (framed in red, 50 µL droplets). The graphs on the right exemplarily depict frozen fraction ($f_{ice}$) values measured with both instruments and the derived INP concentrations ($N_{INP}$).

In contrast, there is super-isokinetic sampling (sample flow velocity higher than TAS), where particles with a sufficiently large inertia are under-sampled (Brockmann, 2011).

For the design of HERA, the layout and inner diameter of the tubing leading up to the filter surface needed to be optimized with respect to the target flow rate and pressure regime to minimize gravitational settling and impaction of supermicron particles due to inertia. Figure 2 shows the transmission efficiency of particles in HERA in a size range from 0 to 20 $\mu$m for different volumetric flow rates ranging from 5 to 100 L min$^{-1}$ at two pressure levels, 1013 mbar and 200 mbar. Calculations were performed with the Particle Loss Calculator (von der Weiden et al., 2009), assuming spherical particles with a density of
2 g cm$^{-1}$ and a temperature of 20 °C. The inner tube diameter leading up to the filter holders is 16.57 mm. Note that these calculations only include the transmission efficiency from the HERA inlet to the filter surface. Neither the aspiration efficiency, nor particle losses in the aircraft inlet and tubing leading up to HERA are included, as they are specific to different campaign setups. It can be seen that there is a strong dependency of the transmission efficiency on the flow rate, with lower flow rates causing fewer losses of supermicron particles due to reduced impaction in the bends. An exception are very low flow rates
≤5 L min$^{-1}$, where the lower flow velocity causes stronger gravitational settling in comparison to sampling at 10 L min$^{-1}$. At a flow rate of 10 L min$^{-1}$, 50 % of particles with a diameter of 11.4 $\mu$m ($D_{50}$) are transmitted, whereas $D_{50}$ is shifted to 7.0 $\mu$m at 40 L min$^{-1}$. For flow rates larger than ~60 L min$^{-1}$, the flow within HERA becomes turbulent for near-surface





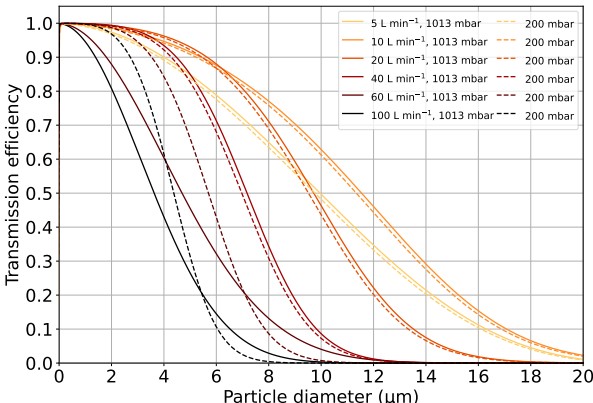

**Figure 2.** Transmission efficiency of spherical particles with a density of 2 g cm$^{-1}$ at 20 °C from the HERA inlet to the filter surface at two different pressure levels (solid lines: 1013 mbar, dashed lines: 200 mbar) in dependence of the volumetric flow rate. Lighter colors mark lower flow rates, darker colors mark higher flow rates.

pressure conditions, leading to a decrease in transmission efficiency for the majority of the particle size distribution. At low pressure, laminar flow conditions can be maintained for flow rates between 60 L min$^{-1}$ and 100 L min$^{-1}$ with $D_{50}$ of 5.7 $\mu$m and 4.4 $\mu$m, respectively. To summarize, the HERA geometry allows for efficient supermicron particle sampling over a wide range of flow rates and pressure levels. Substantial losses are only to be expected for particle diameters larger than ~5 $\mu$m at volumetric flow rates $\geq$60 L min$^{-1}$. Diffusional losses are negligible for particles larger 100 nm (transmission efficiency $\geq$99.5 % for the shown range of flow rates and pressures). In any case, the particle loss calculations presented here can be used, together with information about the inlet and tubing layout and simultaneous measurements of the aerosol particle size distributions, to correct the size distribution of particles sampled on the filter.

## 2.3 Operation

Due to the fast ($<$ 1 min) and remote switching between six filters per flight, HERA opens up new possibilities in terms of sampling strategy. Height-resolved sampling below, inside, and above a cloud layer could give insight about the effect of the available INPs on the formation of the cloud. Comparing filters sampled in different air masses or above contrasting surface features might hint towards the source of the INPs. The strategy and flight pattern should be accounting for sampling periods under somewhat constant atmospheric conditions, e.g., staircase ascents or descents with several minutes of flight time in a constant altitude. The measured INP concentration can later be affiliated with these constant conditions which facilitates the interpretation of results as compared to averaging over a range of different conditions (Coluzza et al., 2017).

To evaluate the filters sampled with HERA, offline INP measurement techniques are needed. These are the Leipzig Ice Nucleation Array (LINA), and the Ice Nucleation Droplet Array (INDA). LINA is a cold stage setup, where 90 1 $\mu$L sized droplets of filter washing water are pipetted onto a hydrophobic glass slide situated on a Peltier element (see blue box in





Fig. 1 c). INDA operates with 50 μL sized aliquots in a 96-well PCR (polymerase chain reaction) tray situated in an ethanol bath (see red box in Fig. 1 c). In addition to measurements with filter extracts, INDA can be used with cut out filter media, e.g., from quartz fiber filters, suspended in ultrapure water. Both setups and temperature calibration routines have previously

been described in detail (Chen et al., 2018; Hartmann et al., 2019). The temperature uncertainty is ±0.32 K for LINA and ±0.50 K for INDA (single standard deviation of at least three calibration experiments). The uncertainty of the measured frozen fractions, i.e., the number of frozen droplets divided by the total number of droplets, is given as the 95 % binomial sampling confidence intervals as described by Agresti and Coull (1998). Frozen fraction measurements from LINA and INDA can be combined by calculating the INP concentration, i.e., normalizing the frozen fraction with the volume of sampled air, the volume

of the washing water, and the droplet volume according to Vali (1971). Differences in the total sampling volume translate to differences in the range of measurable INP concentrations (see Fig. 3). Each box shows the measurable INP concentration range for a specific sampling time at a flow rate of 40 L min$^{-1}$ for either LINA (bluish colors) or INDA (reddish colors), derived from the instrument-specific minimum and maximum measurable frozen fractions, droplet volumes, and filter washing water volumes. LINA and INDA together span an INP concentration range of ~4 orders of magnitude which can be seen when

comparing the upper and lower limits of boxes with the same line style. A shift from the low to the high temperature regime, and with that from the high to the low INP concentration regime, occurs with an increase in sampling time, i.e., sampling volume. Nonetheless, apparently already a very short sampling time of 1 min (solid line) is sufficient to capture high-temperature INPs with INDA if present at concentrations of more than 0.03 L$^{-1}$. However, a very small number of INPs per filter is related to a large statistical uncertainty, while longer sampling times increase the number of INPs per filter and produce data with a higher

statistical significance. Furthermore, one must take the measurement background into account. For INDA, this background is negligible at -10 °C but increases to ~5 INPs per rinsed filter at -20 °C. As a consequence, a sampling time of at least 10 min (dashed line) is necessary to collect a sufficient number of INPs on the filter for INDA measurements above the background at -20 °C. The high INP concentration regime at temperatures below -20 °C can be investigated with LINA and further expanded towards lower temperatures by dilution of the filter extracts with ultrapure water as far as the background of the instrument

allows.

After a research flight, the HERA sampling unit is disconnected from the inlet sampling line and the exhaust line to the pump unit. Only the filter inset is removed from the aircraft and sealed for transport to the laboratory. Filters are removed from their holders under a laminar flow hood, packaged in petri dishes (Analyslide®, PALL cooperation, USA), and kept frozen at -20 °C until used for INP measurements. At least one of the six filters is reserved as a blank, i.e., a filter which is handled in the same

way as the others but is not sampled. This procedure ensures that contaminations are registered and provides a flight-specific background level against which the INP spectra of the corresponding filter samples can be compared. The filter holders are cleaned after each flight in an ultrasonic bath in ultrapure water with a low percentage of ethanol and dried with pressurized, filtered air. Common guidelines for INP-specific filter handling, storage, and measurements are taken into consideration (Polen et al., 2018; Beall et al., 2020; Barry et al., 2021a).





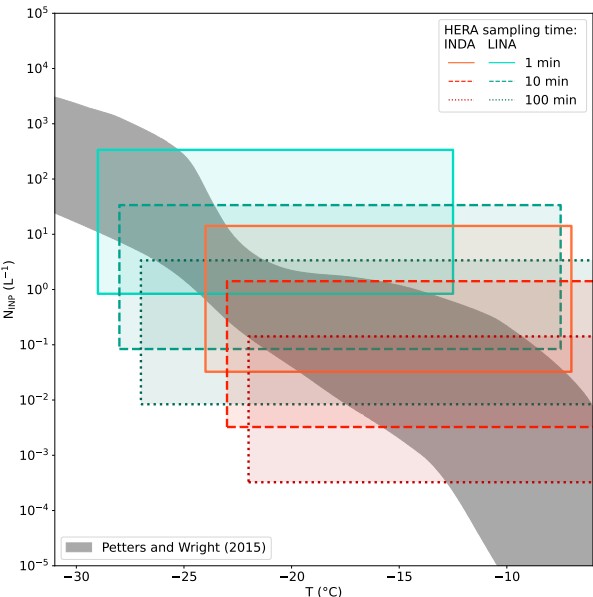

**Figure 3.** Measurable INP concentrations ($N_{INP}$) of INDA (reddish colors) and LINA (bluish colors) when operated with HERA filter extracts. Different sampling times between 1 min (solid lines) and 100 min (dotted lines) with a flow rate of 40 L min$^{-1}$ were assumed. Washing water volumes of 6.2 mL (INDA) and 3.0 mL (LINA) and droplet volumes of 50 $\mu$L (INDA) and 1 $\mu$L (LINA) were considered. Note that due to background effects, INDA and LINA are limited towards low temperatures which is approximated by the left margins of the drawn boxes. The limits towards high temperatures (right margins) are approximated from the intersections with the upper limit of atmospheric INP concentrations derived from precipitation samples in mid-latitudes (grey area in the background; Petters and Wright, 2015).

## 3 Characterization experiments

### 3.1 Effect of filter pore size on INP sampling

Collection efficiencies of PC filters have frequently been measured (Spurny and Lodge, 1972; Burton et al., 2007; Zíková et al., 2015; Soo et al., 2016). For example, 400 nm pore size filters have proven to collect more than 98 % of aerosol particles with diameters between 10.4 nm and 412 nm across a range of flow rates varying between 1.7 L min$^{-1}$ and 11.2 L min$^{-1}$ (Soo et al., 2016). An even higher sampling efficiency is to be expected for filters with a pore size of 200 nm, which are often used for ground-based INP sampling at flow rates below 30 L min$^{-1}$ (DeMott et al., 2016; Knackstedt et al., 2018; Tobo et al., 2019; Tatzelt et al., 2022). However, pre-tests with this filter type have resulted in structural damage of the filter material at 40 L min$^{-1}$ and low pressure (200 mbar), which is why the use of larger pore size filters was considered for HERA.

To investigate the efficiency of 800 nm pore size filters in the context of INP sampling, we used two of the TROPOS-built High-volume And Light-weight Filter samplers for BAlloon-borne appliCation (HALFBAC) and equipped one with a 200 nm and the other one with a 800 nm pore size filter. HALFBAC consists of a filter holder (47 mm, PFA, Savillex, MN, USA), a





vacuum scroll pump (same as in HERA pump unit), temperature, pressure, and relative humidity sensors, radio antenna, GPS module, data logger, and a set of lithium polymer batteries, all contained in a weatherproof housing and weighing below 4.5 kg. The flow rate in HALFBAC is not actively controlled but adjusted via the pump speed prior to sampling while measuring with an external flow meter. Flow rates during sampling are recorded indirectly in the form of differential pressure within a capillary downstream of the filter holder.

Firstly, filters were sampled with polydisperse Arizona Test Dust particles (ATD, nominal fraction 0-3 $\mu$m, Powder Technology Inc., USA) generated from a suspension with an atomizer (similar to 3076, TSI Inc., USA). The suspension was produced by mixing 2.6 g ATD in 50 mL ultrapure water (MilliQ, 18.2 MΩ cm$^{-1}$) and shaking for 15 min. After a settling time of 5 min, the top half of the initial suspension was decanted for further use. The two HALFBACs were connected to the aerosol sampling line to deposit particles onto both filter types in parallel at a volumetric flow rate at the inlet of 15 L min$^{-1}$ generated by the built-in scroll pumps. Secondly, the two HALFBACs were used to sample urban-influenced, continental air on the roof of the Cloud Laboratory at TROPOS in Leipzig, Germany, on February 22nd, 2021. Note that for the ambient sampling, both HALF-BACs sampled through their individual inlets which were pointed into the main wind direction. These filters were sampled simultaneously for 30 min at a volumetric flow rate of 15 L min$^{-1}$. Prior to sampling, the filter holders were cleaned according to the protocol described above (see Sec. 2.3). Filter treatments described in the literature (Barry et al., 2021a) did not lower the measurement background of LINA and INDA which is why filters were used as provided by the manufacturer here and in the following experiments. Post sampling, filters were removed from the HALFBAC filter holders, placed into a centrifuge tube (50 mL, Greiner Bio-One GmbH, Germany) together with 3 mL of ultrapure water and agitated with a laboratory flask shaker for 15 min to wash off collected particles. Frozen fractions were measured with both LINA and INDA. For LINA, droplets were pipetted onto a hydrophobic glass slide (Paul Marienfeld GmbH & Co. KG, Germany). For INDA, another 3.1 mL of ultrapure water had to be added to supply a sufficient sample volume for the 96-well PCR tray (Brand GmbH & Co. KG, Germany). The number of INPs per filter was calculated from the frozen fractions by normalizing with the volume of the washing water and the droplet volume.

The number of INPs per filter with respect to temperature can be seen in Fig. 4. Note that here and in the following, error bars in y-direction only represent the uncertainty of the immersion freezing measurements as described in Sec. 2.3 which is significantly larger than the maximum error in sampling volume (see Sec. 2.1). Error bars are only shown for every fifth data point for better clarity. In case of the polydisperse ATD particles (left panel), the measured number of INPs per filter is independent of the filter pore size. The slight differences in the number of INPs observed at a temperature above -18 °C are within measurement uncertainty. Also in case of the ambient aerosol particles (right panel), both filter types apparently collected comparable numbers of INPs. However, the agreement is much better for the INDA measurements above -18 °C than for the LINA measurements at lower temperatures. The steeper slope of the INP spectrum of the 800 nm pore size filter in comparison to the 200 nm pore size filter below -18 °C is unresolved, but could stem from differences in aspiration efficiency due to the lack of a common inlet. However, it seems unlikely that only low temperature INPs would be affected by this. The described deviation is definitely not related to a lower sampling efficiency of the 800 nm pore size filters, since the number of collected INPs on this filter type is higher in comparison to the 200 nm pore size filter below -21 °C. The statement that





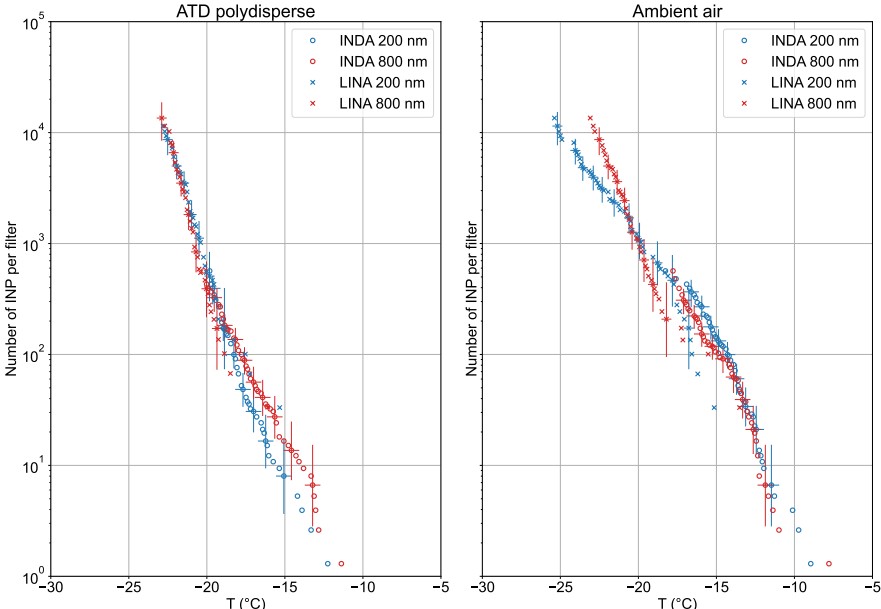

**Figure 4.** Number of INPs detected on filters sampled with polydisperse ATD (left) and urban ambient air (right). Filters with 200 (blue) and 800 nm pore size (red) were sampled simultaneously in two HALFBACs and analyzed with INDA (circles) and LINA (crosses). Note that a common inlet was used for ATD particle sampling whereas each filter was sampled through an individual inlet for ambient aerosol particle sampling. The volumetric flow rate at the HALFBAC inlet was 15 L min$^{-1}$ in all cases.

800 nm pore size filters are just as well suited to collect atmospheric INPs as 200 nm pore size filters is further supported by the fact that the INP numbers agree within measurement uncertainty for polydisperse ATD particles and ambient particles above -21 °C. Our results coincide with measurements by Lacher et al. (2023), who also present comparable results of INP measurements with 200 nm and 800 nm pore size filters from identical sampling periods. Note that an increase in flow rate would even lead to an improved filter efficiency over all particle sizes (Zíková et al., 2015; Soo et al., 2016). Based on these measurements, all of the following results were retrieved using PC filters with 800 nm pore size for particle sampling.

### 3.2 Deposition efficiency of size-selected standard INPs

To characterize the sampling efficiency of INPs with HERA for different particle sizes and flow rates, laboratory experiments with test substances were performed. Briefly, aerosol particles were generated from a suspension with an atomizer, dried, size-selected by sending them through a neutralizer and Differential Mobility Analyzer (DMA, Vienna type, medium), mixed with particle-free, pressurized air to increase the flow rate, and sampled onto filters with HERA. The number concentration of the particles in the sampled air was registered with a condensation particle counter (3010, TSI Inc., USA). Together with the electrical mobility diameter set at the DMA and the sampling flow rate set at the HERA pump unit, the particle surface area and mass per filter could be determined assuming spherical particles.





Two substances, SNOMAX® (SMI Snow Makers AG, Switzerland) and ATD were used for particle generation to investigate the sampling efficiency of INPs of both biological and mineral origin at near-standard pressure conditions. SNOMAX® is a commercially available freezing catalyst containing nonviable cells and fragments of *Pseudomonas syringae* bacteria. The SNOMAX® suspension was generated by dissolving 0.1 g in 50 mL ultrapure water. The ATD suspension was generated in

the same way as described in Sec. 3.1. In total, three different particle sizes (300 nm, 500 nm, 800 nm) were sampled at three different flow rates (10 L min$^{-1}$, 40 L min$^{-1}$, 100 L min$^{-1}$) for both substances. Blank filters were placed in the HERA sampling unit to investigate potential cross contamination due to leak currents through other filter positions (see Appendix A). The positions of both blank and sampled filters in the HERA sampling unit were rotated between trials so that each of the six positions was used equally in the course of the sampling experiments. For the immersion freezing experiments, the SNOMAX®

filters were rinsed with 6 mL ultrapure water, then five 10-fold dilutions of the original extract were produced and investigated with INDA. The ATD filters were treated in the same way as for the filter pore size experiments described above and investigated with both LINA and INDA. Measured frozen fractions of the SNOMAX® filters were normalized with the volume of the washing water, the droplet volume, and the particle mass per filter (density 1.35 g cm$^{-3}$; Wex et al., 2015) to retrieve the ice nucleation active site density per unit mass $n_\mathrm{m}$. In case of ATD, the normalization was performed using the particle surface

area per filter to retrieve the ice nucleation active surface site density $n_\mathrm{s}$.

Figure 5 shows the results of the filter sampling and immersion freezing measurements with monodisperse SNOMAX® particles of different monodisperse diameters sampled at different flow rates with HERA. Each $n_\mathrm{m}$ spectrum is made up of six individual INDA measurements of subsequent 10-fold dilutions of the original filter extract. Of these combined $n_\mathrm{m}$ spectra, each is shown twice: Firstly, in the top row to view potential effects of the flow rate on the sampling of differently

sized monodisperse particles with diameter $D_\mathrm{p}$. Secondly, in the bottom row to compare filters sampled with differently sized monodisperse particles at a constant flow rate $Q$. Overall, we observe good agreement of the $n_\mathrm{m}$ spectra of experiments with different particle diameters and flow rates. For example, the 300 nm particles sampled at a flow rate of 10 L min$^{-1}$ yield similar $n_\mathrm{m}$ values as the 300 nm particles sampled at a flow rate of 100 L min$^{-1}$. The 300 nm particles sampled at a flow rate of 100 L min$^{-1}$, in turn, yield similar $n_\mathrm{m}$ values as the 800 nm particles sampled at a flow rate of 100 L min$^{-1}$. These results

indicate that the ice nucleation active bacteria in SNOMAX® are sampled efficiently with HERA in the investigated range of particle diameters and flow rates. Significant particle losses and/or leaks would lead to a particle-size- or flow-rate-dependent decrease in $n_\mathrm{m}$ which we did not observe. This finding is in line with the results of the particle loss calculations (see Sec. 2.2) that show a minimum transmission efficiency in the here investigated parameter space of 96.4 % (800 nm particles sampled at a flow rate of 100 L min$^{-1}$).

On another note, Polen et al. (2016) describe a decrease in ice nucleation efficiency of SNOMAX® over time, even if the sample was continuously stored at -20 °C. This is significant, since our SNOMAX® batch was more than three years old when the sampling experiments took place. We hence chose to compare the $n_\mathrm{m}$ values from the HERA sampling with measurements from Polen et al. (2016) of a batch that was roughly one year old instead of comparing to a fresh batch. Data by Polen et al. (2016) were generated by producing 0.1 μL droplets from SNOMAX® suspensions with different concentrations and cooling

them down in an oil matrix. The here presented $n_\mathrm{m}$ data lie within the envelope of measurements with "old" SNOMAX®



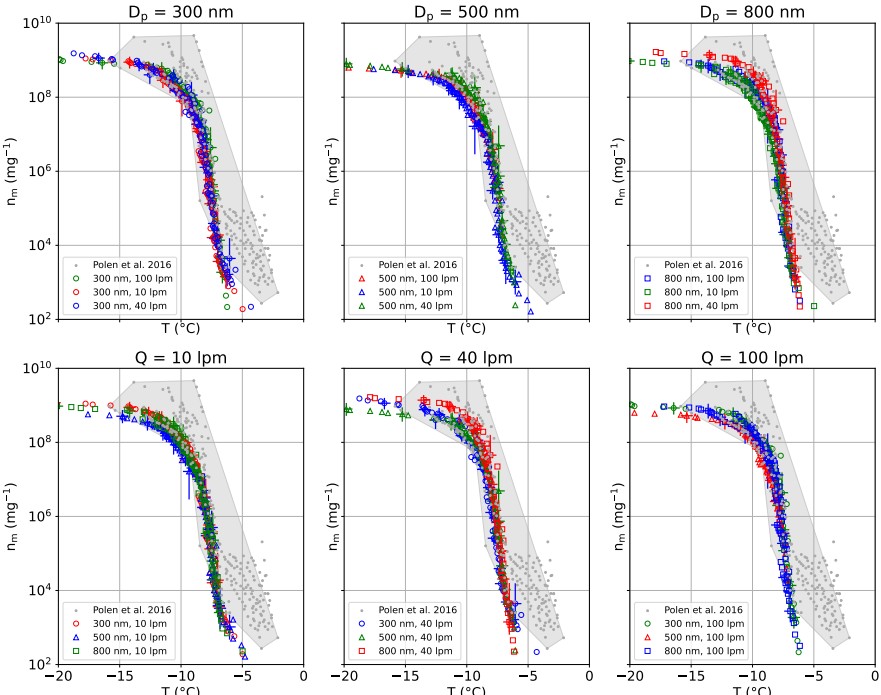

**Figure 5.** Ice nucleation active site density per unit mass $n_{\mathrm{m}}$ with respect to temperature for monodisperse SNOMAX® particles of different diameters sampled at different flow rates with HERA. Each panel in the top row shows results for one particle diameter ($D_{\mathrm{p}}$, circles: 300 nm, triangles: 500 nm, squares: 800 nm) at three different flow rates ($Q$, red: 10 L min$^{-1}$, blue: 40 L min$^{-1}$, green: 100 L min$^{-1}$). Each panel in the bottom row shows results for one flow rate and three different particle diameters (colors and marker shapes as in top row). Data framed in grey in the background were extracted from Polen et al. (2016) using a plot digitizer (Rohatgi, 2022).

by Polen et al. (2016; grey background in Fig. 5). It is not surprising that we do not observe the freezing mode above -5 °C reported by Polen et al. (2016) and in earlier studies (Maki et al., 1974; Yankovsky et al., 1981; Turner et al., 1990; Budke and Koop, 2015). This mode is associated with the occurrence of large aggregates of ice nucleation active proteins which are found in the outer membranes of the *P. syringae* bacteria (Lindow, 1995; Schmid et al., 1997). Bacterial cells have been shown to be

broken up into fragments when spraying a SNOMAX® suspension with an atomizer, significantly reducing the probability of large protein aggregates being deposited on the filters (Wex et al., 2015). The agreement with data from the literature supports the earlier statement of efficient INP sampling with HERA in the investigated parameter space.

    Monodisperse ATD particles were sampled equivalently to the SNOMAX® experiments but were investigated with both LINA and INDA, foregoing the dilution series. It is interesting to note that following the sampling experiments with SNO-

MAX® , the particle generation setup and HERA had to be thoroughly cleaned twice before no more SNOMAX® signatures were observed in the immersion freezing measurements. Figure 6 shows the results of the immersion freezing experiments with 300 nm, 500 nm, and 800 nm particles sampled at 10 L min$^{-1}$, 40 L min$^{-1}$, and 100 L min$^{-1}$. In contrast to the SNOMAX®





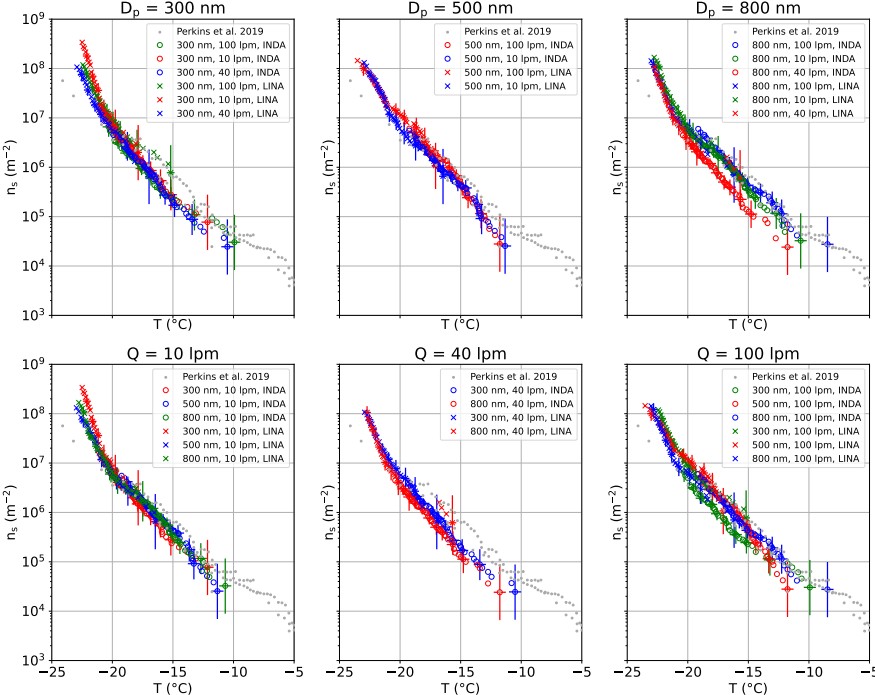

**Figure 6.** Ice nucleation active surface site density $n_s$ with respect to temperature for monodisperse ATD particles of different diameters sampled at different flow rates with HERA. Circles show data retrieved from INDA measurements, crosses data from LINA measurements. Each panel in the top row shows results for one particle diameter at three different flow rates. Each panel in the bottom row shows results for one flow rate and three different particle diameters (colors as in top row). The grey data points in the background were extracted from Perkins et al. (2019) using a plot digitizer (Rohatgi, 2022).

results, $n_s$ was calculated from the total particle surface area of ATD per filter, which is a better measure of ice nucleation efficiency than $n_m$ in case of insoluble materials such as mineral dust (Connolly et al., 2009). Again, each $n_s$ spectrum is shown

twice for better visualization of potential effects of particle diameter and flow rate on $n_s$. Equally to the SNOMAX® results, the ATD $n_s$ spectra are similar to one another in their shape and magnitude. It appears that the spread between the different experiments is highest for the largest particle diameter (800 nm) and the highest flow rate (100 L min$^{-1}$). However, even in these cases, most data points are found within the range of measurement uncertainty and no clear trend in the magnitude of $n_s$ with particle size or flow rate is observed. Furthermore, our results agree well with data by Perkins et al. (2019) who measured

the immersion freezing behavior of 50 $\mu$L aliquots of ATD suspensions with different concentrations with a PCR-tray-based system. In conclusion, HERA is suited for representative sampling of submicron ATD particles for subsequent offline analysis of their immersion freezing behavior.





### 3.3 Deposition efficiency of atmospheric INPs

In addition to the experiments with conditioned particles in the laboratory, atmospheric particles were sampled with HERA to
evaluate the new method for a mixture of particles of different sizes and chemical compositions. HALFBAC was sampling in
parallel to produce a benchmark for the comparison of retrieved INP concentrations. Both instruments were operated on the
roof of the Cloud Laboratory at TROPOS, i.e., at near-standard pressure conditions, with their separate inlets oriented in the
main wind direction. Eight filter samples were collected from each instrument on several days in May, June, and August 2020.
Table 1 lists the date and time of the sampling periods including sampling volume, mean wind speed, and mean temperature as
measured at the TROPOS weather station. Immersion freezing measurements were performed with LINA for filters sampled on
May 28th, 2020 and with INDA for the remaining samples. Figure 7 shows the INP concentrations as determined for the HERA
(squares) and HALFBAC filters (triangles). The top left panel aims to visualize the dependency of instrument agreement on
the meteorological conditions during sampling. Root mean squared logarithmic errors (RMSLE) of INP concentrations from
HERA $N_{\text{INP, HERA}}$ and HALFBAC $N_{\text{INP, HALFBAC}}$ were determined according to Eq. 1 with $n$ the number of available, non-zero
data points:

$$RMSLE = \sqrt{\frac{1}{n}\sum_{i=1}^{n}\Big[\ln(1 + N_{\text{INP,HERA}}) - \ln(1 + N_{\text{INP,HALFBAC}})\Big]^2}. \tag{1}$$

RMSLE values are shown on the y-axis and are contrasted with the variability in wind speed and direction during the
different sampling periods. The single standard deviation in wind direction is shown on the x-axis, whereas the single standard
deviation in wind speed is represented by the marker size. This analysis was performed because HERA and HALFBAC were
not sampling from a common inlet and the aspiration efficiency of particles depends strongly on the wind speed and alignment
of the inlet with the wind direction (see Sec. 2.2). Overall, instrument agreement seems to decrease with an increase in wind
direction variability, whereas no clear dependency on wind speed variability can be found for the eight sampling periods.
Moreover, the RMSLE values are apparently not correlated with the average temperature or wind speed during sampling (see
Tab. 1). Grey data points indicate filter samples that were collected during periods with variable wind direction (single standard
deviation $\geq 30°$), which are not shown in the remaining panels of Fig. 7. Concerning the HERA–HALFBAC comparison
during the remaining five sampling periods with steady wind direction, we find very good agreement between the two samplers
in both shape and magnitude of the INP spectra (see top right and bottom panels of Fig. 7 with the colors corresponding to
those in the top left panel). From this, we conclude that the two instruments feature a similar INP sampling efficiency as long
as the aspiration efficiency is comparable.

## 4  First results from aircraft sampling

As described in Sec. 1, a prototype of HERA was deployed on the Polar 5 aircraft of the Alfred Wegener Institute (AWI)
during PAMARCMiP 2018. For this first test, only one filter was sampled per flight at a flow rate of 10 L min$^{-1}$ (Hartmann





**Table 1.** Sampling periods for the comparison of INP concentrations from filters sampled with HERA and HALFBAC. The meteorological data was measured at the TROPOS weather station and averaged over the sampling period.

| sample ID | date | time start (UTC) | time stop (UTC) | volume (L) | temperature (°C) | wind speed (m s$^{-1}$) |
|-----------|------|------------------|-----------------|------------|------------------|-------------------------|
| 1 | 2020-05-28 | 09:03:00 AM | 11:33:00 AM | 4500 | 15.8 | 3.8 |
| 2 | 2020-05-28 | 12:54:00 PM | 03:24:00 PM | 4500 | 18.2 | 3.2 |
| 3 | 2020-06-02 | 07:40:00 AM | 10:10:00 AM | 4500 | 21.7 | 1.4 |
| 4 | 2020-06-02 | 10:52:00 AM | 01:22:00 PM | 4500 | 24.7 | 1.7 |
| 5 | 2020-08-06 | 12:25:00 PM | 03:05:00 PM | 4800 | 30.6 | 1.7 |
| 6 | 2020-08-11 | 08:15:00 AM | 10:44:00 AM | 4470 | 29.6 | 2.6 |
| 7 | 2020-08-11 | 12:45:00 PM | 02:15:00 PM | 2700 | 32.1 | 2.6 |
| 8 | 2020-08-12 | 08:32:00 AM | 11:02:00 AM | 4500 | 29.8 | 3.0 |

et al., 2020). Only afterwards, HERA was equipped with the pump unit described earlier (see Sec. 2.1) to achieve higher flow rates. The first application of the upgraded HERA system was the HALO mission CIRRUS-HL in June and July 2021. For

this, HERA was installed on HALO as shown in Fig. 1, with sampling lines from both the HASI for sampling outside cloud aerosol particles and the CVI inlet for sampling cloud particle residuals. The HASI and the aerosol sampling line were revised prior to CIRRUS-HL in cooperation with enviscope GmbH to enable more efficient sampling of supermicron aerosol particles at flow rates larger than 30 L min$^{-1}$. Briefly, the setup was changed from several small diffusors within a main diffusor each being connected to the instruments with their individual sampling lines (Minikin et al., 2017), to a single diffusor connected

to a main sampling line with larger inner diameter. All instruments were connected to this main sampling line at individual junction points, with HERA sampling at the end of the line at a volumetric flow rate of 40 L min$^{-1}$. The total airflow was regulated according to the TAS via a bypass to ensure near-isokinetic sampling at all times. Furthermore, compensation pumps were installed to minimize variations in the total flow rate when the HERA pumps were not running. The volumetric flow rate at the CVI was ~5 L min$^{-1}$.

The HERA filters from CIRRUS-HL were removed from the sampling unit, packaged, and stored until evaluation with LINA as described in Sec. 2.3. Instead of hydrophobic glass slides, we used Si wafers (100 orientation, undoped, 50.8 mm, Si-Mat Silicon Materials, Germany) as a substrate. The temperature of the droplets on the Si wafers was calibrated using higher alkanes (n-undecane and n-tridecane, 99 %, Thermo Fisher Scientific Inc., USA) with defined melting points as described by Budke and Koop (2015). The temperature uncertainty was estimated to be ±0.33 K (single standard deviation of three individual

measurements with both substances). In test measurements with ultrapure water, frozen fractions of an ensemble of 1 $\mu$L sized droplets tended to be significantly shifted towards lower temperatures when comparing the Si wafers to the hydrophobic glass slides. On average, a decrease of the temperature of a frozen fraction of 50 % of 3 K was observed when using Si wafers instead of the glass substrates. As a consequence, Si wafers were chosen over hydrophobic glass slides as the standard substrate for the CIRRUS-HL filter evaluation. In contrast to the LINA measurements presented in the previous chapter, measurements were





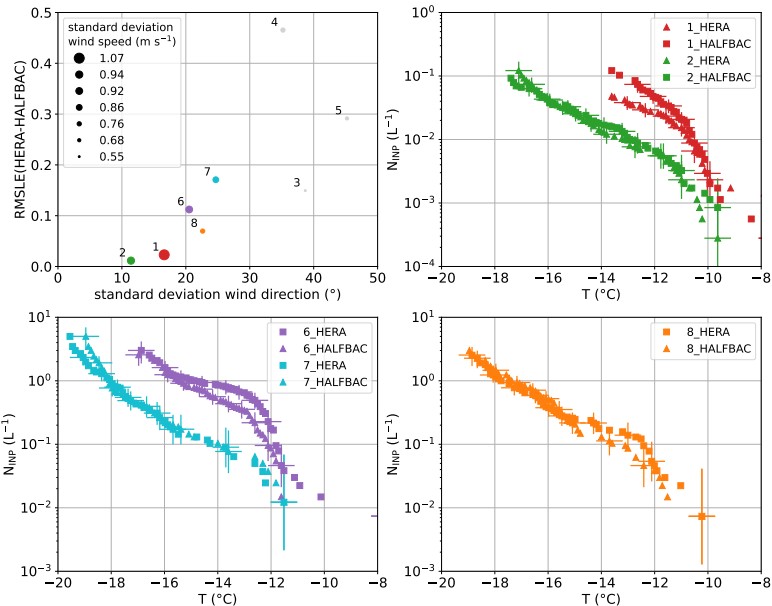

**Figure 7.** INP concentrations ($N_{INP}$) from filter samples collected with HERA (squares) and HALFBAC (triangles) with respect to temperature. The top left panel shows the RMSLE-based deviation between INP concentrations from the HERA and HALFBAC filters with respect to the variability in wind direction during the sampling periods. Note that for this, only the temperature range was used where non-zero data from both instruments were available. The marker size refers to the variability in wind speed. Colored data points correspond to the INP spectra shown in the remaining three panels. Grey data points indicate sampling periods with high variability in wind direction for which the INP spectra were not further analyzed.

performed with 55 droplets instead of 90. This is due to the surface properties of the Si wafers for which droplets feature a smaller contact angle and thus spread out over a larger area in comparison to the hydrophobic glass slides. The total number of droplets hence had to be decreased to 55 to still fit on the cooling element of LINA.

Figure 8 shows frozen fractions with respect to temperature retrieved from three sampled filters and one blank of research flight (RF) 15 of CIRRUS-HL on July 13, 2021. The filters sampled at the HASI are shown in red and green, the filter sampled

at the CVI in blue, and the blank in grey. The HERA inlet pressure during sampling ranged between 1030 and 220 mbar mbar. The average cabin pressure was ~800 mbar. The background of the ultrapure water (light blue area in Fig. 8) represents the upper and lower limits including uncertainty from six measurements on Si wafers of the same charge as used for the filter samples of RF 15. On the one hand, it can be seen that the blank is close to the ultrapure water background, indicating that only very few additional INPs were introduced due to filter handling and storage in HERA. The filters sampled at the HASI

and CVI, on the other hand, show significantly higher onset freezing temperatures than the blank and the ultrapure water. All





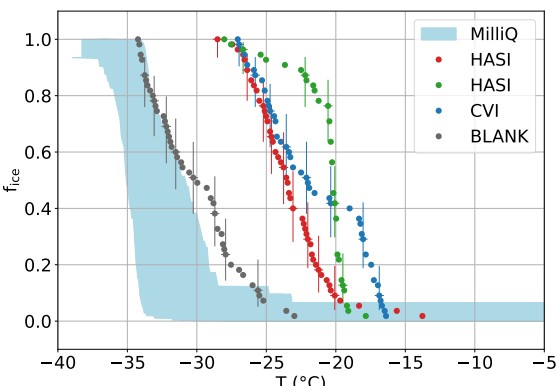

**Figure 8.** Frozen fraction ($f_{ice}$) with respect to temperature from RF 15 of CIRRUS-HL measured with LINA. Aerosol particle filters sampled at the HASI are shown in red and green, a cloud particle residual filter sampled at the CVI in blue, and a blank filter in grey. The ultrapure water background (MilliQ) is shown in light blue in the background and represents the upper and lower limits of six water measurements with Si wafers from the same charge, including the measurement uncertainty.

droplets are frozen at -28 °C while the frozen fraction is only ~30 % in case of the blank filter at the same temperature. The vast majority of INPs sampled on the filters must hence stem from the air collected through the aircraft inlets. The frozen fraction measurements of the HASI and CVI filter samples show distinct features indicating that different INP populations with specific immersion freezing properties have been collected. While the discussion of these features is beyond the scope of this study, the observed differences between samples are suggestive of the sensitivity of the HERA filter samples with respect to variations of atmospheric INP concentrations. To summarize, also at low in-line pressure there is neither a noticeable cross-contamination betwenn the HERA filter samples nor a significant contamination from filter handling or leaks between HERA and the pressurized cabin.

## 5 Summary and outlook

In this paper we introduced the new High-volume Flow Aerosol Particle Filter Sampler (HERA) for aircraft application. HERA can be equipped with up to six filters, with in-flight filter changes realized with the help of an electrically driven valve. The powerful, actively-controlled pump unit enables sampling at a flow rate exceeding $100\,\mathrm{L\,min^{-1}}$, depending on the filter medium and pressure conditions. The system was designed for efficient sampling of supermicron particles at high flow rates ($D_{50}$ = $7\,\mu$m at 40 L min$^{-1}$). These features make HERA highly automatable, minimize the risk of contamination, and enable high temporal and spatial resolution of INP concentration measurements.

Proof of principle experiments with SNOMAX® and ATD were conducted. For this, particles were generated from a suspension, size-selected, and sampled at different flow rates onto filters with HERA, followed by rinsing the filters to generate a suspension for immersion freezing experiments. Monodisperse particles between 300 nm and 800 nm were generated and



sampled at flow rates between 10 L min$^{-1}$ and 100 L min$^{-1}$ at near-standard pressure. We did observe good agreement of
the ice nucleation active site density per SNOMAX® mass and ATD surface area in comparison to literature results (Polen
et al., 2016; Perkins et al., 2019), where suspensions were directly used for immersion freezing measurements. Furthermore,
no dependency of particle size or flow rate on the results of the immersion freezing experiments was found. These findings
suggest efficient sampling of INPs without any alteration of their immersion freezing properties with HERA in the investigated
parameter space.

The performance of HERA was compared to the more straightforward filter sampler HALFBAC by ground-based collection
of atmospheric aerosol particles and analysis of their immersion freezing behavior. A correlation of the difference in INP
concentration from HERA and HALFBAC with the variability in wind direction during the sampling periods was found. This
dependency was interpreted as being due to the lack of a common inlet and associated differences in aspiration efficiency.
Filters from both instruments yielded similar results as long as the wind speed and direction during the sampling period were
stable.

During the CIRRUS-HL mission, HERA was operated on HALO for the first time. Results from RF 15, where three filters
were sampled at HERA inlet pressure values between 1030 mbar and 220 mbar, show a blank filter background close to the
the ultrapure water. The filters sampled at the HASI and CVI inlets each featured distinct freezing spectra and ice nucleation
activity significantly above the blank background. These results indicate the sensitivity of the immersion freezing measurements
of the HERA filter samples with respect to different atmospheric conditions. Furthermore, it can be concluded that no notable
contamination was introduced via filter handling and leakage currents between filter holder pathways in HERA or between
HERA and the pressurized aircraft cabin.

Future investigations will focus on the evaluation of the HERA filter samples from the HALO CIRRUS-HL mission with
respect to the origin of the sampled air masses, aerosol particle and cloud particle residual size distributions, and chemical
composition. In addition, HERA was operated on the AWI Polar 6 aircraft during HALO-(AC)[3] (Arctic Air Mass Transfor-
mations During Warm Air Intrusions And Marine Cold Air Outbreaks) in spring 2022 and BACSAM I (Boundary Layer and
Atmospheric Aerosol- and Cloud Study) in fall 2022 and data are currently being evaluated. In the future, we plan to investigate
HERA filter extracts with alternative offline methods featuring lower background levels, e.g., microfluidics (Stan et al., 2009;
Reicher et al., 2018; Tarn et al., 2018), to increase the measurable INP concentration range. Generally, HERA filters could
also be investigated with other techniques such as scanning electron microscopy for particle morphology analysis (Sanchez-
Marroquin et al., 2021; Seifried et al., 2021) or ion chromatography for bulk chemical composition analysis (Kwiezinski et al.,
2021).

So far, filter changes in HERA have been triggered by an on-board operator. For more automated sampling in the future, it
could be feasible to use information from other systems on the aircraft (geographical position, altitude, temperature, pressure,
or others) as input parameters for the HERA software and trigger filter changes according to certain threshold values. This
would enable the application of HERA on a more regular basis, e.g., on commercial aircraft or measurement campaigns with a
very limited number of on-board operators. Further HERA systems could be produced for simultaneous integration on different





aircraft. With this, the currently small set of free tropospheric INP concentration data could be expanded to further improve our understanding of the role of INPs on cloud formation and properties.

*Data availability.* The datasets will be made available on Zenodo.org in the near future. Furthermore, data are available upon request to the contact author.

## Appendix A:  Quantification of leakage currents

During the construction of HERA, care was taken to avoid leaks. The previously presented results of the sampling and immersion freezing experiments with SNOMAX® and ATD (see Sec. 3.2), as well as the comparison of the performance of HERA

and HALFBAC (see Sec. 3.3) suggest no relevant leaks at near-standard pressure. Nonetheless, leakage currents between the filter holder pathways in the seven-way valves were determined to provide supporting evidence. In principle, two types of leaks are possible which could lead to contamination of the filter samples and/or an overestimation of the total sampling volume per filter in HERA. Firstly, there could be leaks between the filter holder pathways leading to cross-contamination. Secondly, there could be leaks between HERA and the ambient environment leading to contamination and/or an error in total sampling volume

depending on the position of the leak.

During the above described sampling experiments with SNOMAX®, blank filters were placed in different positions of HERA while another filter was sampled. The blanks were analyzed with INDA in the same way as the sampled filters and showed ice nucleation activity above the ultrapure water background (see left panel of Fig. A1; example for 500 nm SNO-MAX® particles sampled at 40 L min$^{-1}$). This is to be expected, since the sampled aerosol contained several orders of

magnitude more INPs than found in the atmosphere, and thus even very low leakage currents will lead to SNOMAX® particles being deposited on the blanks. Average particle number concentrations ranged between 300 cm$^{-3}$ and 3 cm$^{-3}$ during sampling of 300 nm and 800 nm particles, respectively. According to Hartmann et al. (2013) it can be estimated that ~3 % of the 300 nm and ~40 % of the 800 nm particles contained at least one ice nucleation active protein aggregate. With this, the INP concentration in the sampled air during our laboratory experiments was 3 to 4 orders of magnitude higher than the maximum

atmospheric INP concentration at -15 °C (Petters and Wright, 2015). We made use of the immersion freezing results of the blanks to calculate the total volume of air that must have flowed through the blank filters. In detail, the INP concentration of the blank filter was fit to the INP concentration of the corresponding filter sample by adjusting the sampling volume through the blank until the logarithmic least squares error was minimal. Figure A1 (right panel) exemplarily shows the procedure for the sampling of 500 nm SNOMAX® particles at 40 L min$^{-1}$. The blue data points show the INP concentration of the filter which

was sampled with a total of 6000 L, the red data points the INP concentration of the blank filter assuming a sampling volume of 10 L as a first guess. It can be seen that this first guess is ~2 orders of magnitude too high, or, in terms of concentrations, would produce values which are ~2 orders of magnitude too low. Fitting the blank data to the sampled filter data (red line in Fig. A1) results in a much lower sampling volume through the blank of 78.6 ± 1.8 mL, or in terms of leakage current, 0.52 ± 0.01 mL



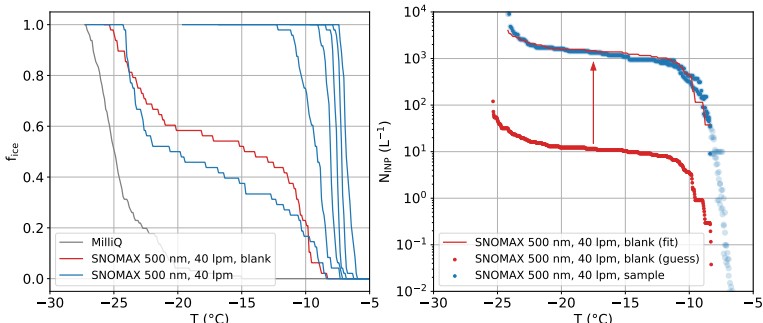

**Figure A1.** Left: Frozen fractions with respect to temperature from the dilution series of the filter sampled with 500 nm SNOMAX® particles at a flow rate of 40 L min$^{-1}$ (blue), the corresponding blank (red), and an ultrapure water measurement from the same day (MilliQ, grey). Right: INP concentration with respect to temperature as determined from sampling of 500 nm SNOMAX® particles at a flow rate of 40 L min$^{-1}$ (blue). The red data points are the first guess of the INP concentration of the blank (10 L sampling volume). The red line indicates the fit function of the blank filter after adjusting the total volume of air to best match the data of the corresponding sampled filter. A leakage current of $0.52 \pm 0.01$ mL min$^{-1}$ was detected. Data outside the blank measurement range, which were not used for fitting, are shown in light blue.

min$^{-1}$ (~0.001 % of total flow rate). The procedure described above was performed for all blank filters collected during the

SNOMAX® sampling utilizing the six positions and all were within 0.2 mL min$^{-1}$ of the result presented in Fig. A1. In accordance with the immersion freezing results of the filters sampled with SNOMAX® and ATD (see Sec. 3.2), no dependency of the particle size or flow rate on the leakage currents could be observed. During a sampling time of five hours, in which 12,000 L would be sampled at a flow rate of 40 L min$^{-1}$, the determined leakage currents would generate a volume of ~0.2 L through other filter positions, which translates to a contamination of less than 0.4 INP per filter (assuming a maximum INP

concentration of 2 L$^{-1}$ at -20 °C; Petters and Wright, 2015).

Leaks between HERA and the aircraft cabin are a second potential source of contamination. In case of low in-line pressure and a pressurized aircraft cabin, the risk of aerosol particles entering the system through leaks and depositing on the filters is enhanced in comparison to a non-pressurized cabin. This is avoided by thorough leak testing in the laboratory prior to a campaign, i.e., by evacuating the system and determining its relaxation time and by comparing flow rates measured at the

HERA inlet and in the pump unit at low pressure. Ultimately, potential leaks can be identified during measurement flights (see Sec. 4).

*Author contributions.* The experiments were conceptualized by CJ, HW, and FS. The sampling experiments were performed by CJ, JS, HW, and SG. The immersion freezing experiments were performed by JS and SG. The data evaluation and interpretation was performed by SG with contributions from JS, HW, and FS. SG wrote the manuscript with contributions from all co-authors.



*Competing interests.* The authors declare that they have no conflict of interests.

*Acknowledgements.* We thankfully acknowledge the funding by the Deutsche Forschungsgemeinschaft (DFG, German Research Foundation), projects 316508271 and 442648163 within PP 1294 (HALO). We thank Thomas Conrath and Astrid Hofmann (both TROPOS) for technical support regarding the hard- and software of HERA. We thank Josephine Gundlach and Markus Hartmann (both TROPOS) for support with the immersion freezing experiments and Kerstin Flachowsky (TROPOS) for providing data from the TROPOS weather station.
We further thank Hans-Christian Clemen (MPI-C, Mainz) for the operation of HERA during RF 15 of CIRRUS-HL.



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
