# Peer review of "Next-generation ice nucleating particle sampling on aircraft: Characterization of the High-volume flow aERosol particle filter sAmpler (HERA)"

_Atmospheric Measurement Techniques, 2023_

## Author Comment (AC1)

**Author response to comments by Gabor Vali (Referee #1):**

We thank Gabor Vali for his valuable comments and recommendations to our manuscript. We feel that due to these comments and our respective adjustments, the readability and quality of our manuscript is now improved, but intently anticipate reassessment by the referee.

In the following, we address the referee comments and describe adjustments made. For this, the referee comments are given in blue and our answers in black. When referencing page, line, and section numbers, we always refer to the first version of the manuscript unless otherwise stated.
* * *
The paper presents a newly designed sampling device for atmospheric ice nucleating particles (INPs). It is designed for use in research aircraft. The design is described in detail and test results are presented to show the sampling efficiency and reliability of the device. The significance of the results is the prospect that it will aid in obtaining research results on atmospheric INPs at higher spatial and temporal resolution than has been possible with other systems. This can be specially useful when coupled with cloud and precipitation studies. The paper addresses the relevant questions thoroughly, although with some limitations and with some superfluous material. The topic of the paper fits well within the scope of AMT.

The manuscript presents an evaluation of the HERA sampling unit specifically for INPs. To some extent this is justified as the stated goal for HERA is to collect samples for INP studies. On the other hand, it makes the paper somewhat convoluted and focus is lost on the fact that the novelty is the HERA sampling, not the filter processing. The latter issue has been extensively treated in previous publications. Some new data are included here on pore-size dependence but really that should be a topic examined on its own.

Except for rather special situations, INPs can be expected to be dynamically just like other aerosol particles. Thus, transmission and collection efficiencies for INPs of given sizes, densities and shapes can be expected to be the same as for similar particles of other kinds. In view of that, it would be better if the paper separated the HERA sampling efficiency from INP processing issues. In fact, it is curious why the authors focus only on INPs and do not leave open the use of HERA for aerosol sampling for other types of aerosol analyses.

As the referee points out, the focus of HERA is the sampling of INPs. This is why we chose to validate the collection efficiency via immersion freezing measurements. These involve a number of steps which are specific to the here presented measurements and the used setups and must be described in the manuscript. In order not to distract from the results, we shifted the specifics of the INP analysis to the Appendix, where it is easily available to the reader. The specifics of the sampling remain in the main part of the manuscript. With that, and some additional information concerning sampling efficiencies on aircraft and in the HERA-HALFBAC comparison experiments, we hope to have shifted the focus more towards filter sampling.

We chose the method of validating the collection efficiency of HERA via filter sampling and immersion freezing measurements of the filter washing water, since this is the general use case for HERA. The general aerosol particle population is usually analyzed with the help of online instruments on aircraft, which offer a higher temporal resolution. Of course, there are some exceptions to this. We added the following sentence in the introduction to better clarify the scope of the study: "While the HERA filter samples can be used for a number of different types of aerosol particle analyses, e.g., scanning electron microscopy for particle morphology analysis (Sanchez-Marroquin et al., 2021; Seifried et al., 2021) or ion chromatography for bulk chemical composition analysis (Kwiezinski et al., 2021), this study focuses on the application for immersion INP measurements."

The paper also gets complicated by not clearly defining that installation-dependent other factors, specially inlet characteristics, are not subjects of this paper. Only in the discussion of the flight result (Section 4) do these issues have a place, as a specific instance of information needed to interpret the results.

Indeed, it was our original intention to separate the characterization of HERA itself from installation-dependent factors. However, we also understand the request for information on the transmission efficiency on aircraft, which was also brought forward by referee #2. We performed calculations of transmission efficiency for the CIRRUS-HL setup and added the results in Sec. 4: "[…] Assuming spherical particles with a density of 2 g/cm, an inline pressure of 340 mbar, and an inline temperature of 26 °C (corresponding to ~200 m/s TAS), $D_{50}$ at the HASI is 2.7 µm (aspiration and transmission efficiency). The HALO-CVI sampling line to HERA had a total length of ~5 m (10 mm inner diameter until flow distribution at roughly half of total length, then contraction to 4.57 mm) resulting in $D_{50}$ =2.2 µm for the above given conditions ($D_{50}$ =4.7 µm from HALO-CVI inlet to flow distribution). "

The HERA unit is aimed to facilitate the collection of filter samples for INP analyses by eliminating in-flight handling of the filters. This is a worthwhile goal, although controls used in many previous cases showed that manual exchanges of filters didn't produce noticeable contamination. The potential of HERA will be more fully evident once the operation is fully automated and the need for a technician attending to the collector is eliminated. Then, the remaining limitation will be the number of filters per flight to six. Selection of six truly meaningful sampling intervals will require difficult decisions by the flight directors. The paper could be more realistic about these issues. Also, it should be clarified if filter holders could be exchanged during flights or not. Perhaps this is already in the paper and was missed in reading it.

The referee is correct in mentioning that results from previous aircraft filter sampling do not report noticeable contamination. However, only very few studies mention the collection of field blanks in the first place, and those that do, did only collect them sporadically.  Hence, contamination might not have been detected. To clarify this issue, the respective sentence in the original manuscript was changed to: "While no significant contamination was reported in the above mentioned studies, blanks were not taken during every flight and contamination might have been missed depending on the frequency of occurrence. "

In principle, it would be possible to remove the filter holder inset in flight and replace the filter holders with "fresh" ones. This was added to the manuscript in Sec. 2.3. However, this is not possible on the HALO aircraft, where instrument parts must not be removed in-flight due to safety regulations.

It is correct, that the selection of meaningful sampling intervals is challenging. However, this challenge would prevail even with more available filter holders. To clarify the difficulties, we reformulated the paragraph in question in Sec. 2.3: "Any kind of aircraft INP filter sampling involves careful planning to achieve truly meaningful sampling intervals. In general, the flight pattern should be accounting for sampling periods under somewhat constant atmospheric conditions […]. In practice, the sampling strategy often must be adapted in-flight due to unforeseen changes in weather conditions and/or flight track. Consequently, fast decisions by the on-board operators are needed which can be easily realized with HERA due to the quick (< 30 s) and remote-controlled switching between filters. The number of six filters per flight was based on typical flight durations and expected INP concentrations in the free troposphere and so far was found appropriate in practice. If more filters are needed and the aircraft certification regulations allow for it, the filter holder inset could be removed in-flight and filter holders could be exchanged."

The design of HERA is described in great detail. Yet, one misses information about overall size, weight and power requirements. Those are important factors for aircraft deployment.

We apologize for not mentioning these important facts. We included the following in Sec. 2.1: "The sampling unit measures 49 cm × 52 cm × 27 cm (width × depth × height), fits into a standard 19 inch rack unit, and weighs 22 kg. […]  The pump unit measures 49 cm × 25 cm × 18 cm, weighs 13 kg […]. At full pump speed, the power consumption of HERA is ~400 W ."

Sampling efficiency is presented in terms of theoretical calculations. It would have been good to have support for the computed efficiencies from actual tests. The tests with Snomax are in that direction but for smaller sizes than the calculations (Fig. 2).

We agree with the referee. Sampling of supermicron particles was considered for the experiments with SNOMAX and ATD, but not performed due to the difficulties of supermicron particle generation and size-selection and the related significant increase in sampling times. For the submicron particles, our results fit the calculations of the transmission efficiencies. For now, nothing can be said about the actual transmission of supermicron particles. We added the following in Sec. 3.2: "Due to the particle generation setup, the sampling experiments were restricted to particles with mobility diameters ≤ 800 nm, where particle losses should be minimal according to the theoretical calculations (see Fig. 2, minimum transmission efficiency of 96.4 % for 800 nm particles sampled at a flow rate of 100 L/min and near-standard pressure). Regarding supermicron particles, a decrease in transmission efficiency is expected according to the calculations, but experimental results cannot yet be provided."

A design criteria of 0.5 µm lower limit for particle sizes is given and transmission efficiencies are calculated for size above that limit. Small particles are less subject to losses. This is fine, but the justification given deserves some comments. Two references are cited in support of

the decision for INPs. Both references show correlations of INP concentrations with particle sizes >0.5 μm but also show temperature dependence and variations with ambient conditions. None of the correlations are strong at higher nucleation temperatures and, in any case, correlations can arise from source or transport similarities without the INPs actually being >0.5 μm. Both studies were made with samples at ground level and there are other limitation as well in the two references. It is out of place for this manuscript to review the literature about INP sizes. It should however not give the appearance of a firm justification for the choice. Results obtained with the use of the HERA will have to be evaluated with the size question in mind, specially since the impact of INP sizes is also an issue for INP extraction and detection with the filter method.

The particle size range of > 0.5 μm was merely cited from the literature because correlation between the concentration of these particles and INP concentrations was observed. It was not our intention to communicate 0.5 μm as the lower size limit for particle collection with HERA. Transmission efficiencies were also calculated for particles < 0.5 μm (see Fig. 2). We agree that stating a lower size limit for INPs is not appropriate and decided to leave out any size information in this part of the manuscript to avoid confusion. The text was adapted in the following way: " While there are several factors influencing the potential of an aerosol particle to act as an INP, size seems to be an important one, as the concentration of large particles has been shown to correlate with the INP concentration (Pruppacher and Klett, 1997; DeMott et al., 2010). INP sampling should hence be setup so that losses of large particles are minimized."

With the foregoing in mind, it is recommended that the Introduction focus on the sampling issues from aircraft, that Section 3.1 be re-written to focus on showing the lack of sensitivity to flow-rate for small particles (as expected) for one filter setup. Sections 3.2, 3.3 and 4 are the main results to present. The use of 'deposition efficiency' should be avoided and it is easily confused with deposition nucleation tests. The discussion about cell fragments versus protein aggregates isn't effective because of the differences in physical dimensions; the sources of differences are more likely to be a question of sample aging and other treatment differences. The observed drop-off in detection at the higher temperatures is concerning but not a point to be treated in this paper.

We are not completely sure that we understand the comment concerning the re-writing of the Introduction and Sec. 3.1. The introduction already includes an overview of INP sampling activities and issues on aircraft (p. 3-3, l. 80-101). Section 3.1 deals with the effect of filter pore size on INP sampling at a single flow rate. This experiment did not only focus on "small" particles, but was performed with polydisperse ATD and ambient air, which most definitely included some supermicron particles. We respectfully ask the referee to clarify his request concerning the content of the mentioned sections and we will implement adjustments in the next round of reviews.

The term "deposition efficiency" was replaced with "collection efficiency" at all instances.

Concerning the missing class A mode in the INP spectra of SNOMAX after filter sampling, we believe that our original explanation is a possible cause. The class A protein aggregates can be several tens of nanometers in size (Govindarajan and Lindow, 1988) and might very well be separated when the much larger *P. syringae* cells are fragmented in the atomizer. Wex et al. (2015) report sizes of cell fragments ranging from 50 nm to 250 nm after spraying with an atomizer. From our point of view, this argumentation is quite convincing and we would like to stick with it in the manuscript. However, we adjusted the paragraph slightly to leave more room for interpretation: "Interestingly, we do not observe the freezing mode above -5 °C [...] which can have several causes. This mode is commonly associated with the occurrence of large aggregates of ice nucleation active proteins which are found in the outer membranes of the P. syringae bacteria (Lindow, 1995; Schmid et al., 1997). Bacterial cells have been shown to be broken up into fragments when spraying a SNOMAX® suspension with an atomizer (Wex et al., 2015), reducing the probability of large protein aggregates being deposited on the filters. Another reason for the missing high-temperature mode could be the prolonged storage of the SNOMAX® batch leading to some kind of deactivation of the large protein complexes."

In Section 3.3. the experimental setup would be of interest and the reason for the wind-speed dependence more understandable. This is more to the central point of the paper (sampling efficiency).

We added more detail to the description of the experimental setup in Sec. 3.3, e.g., the inlet inner diameters of the two instruments. In accordance with comments by referee #2, we performed additional calculations of the aspiration efficiency in dependence of wind speed and direction to estimate the effect of these two quantities on the overall sampling efficiency. It turns out that in our experiments due to the smaller inlet diameter of HALFBAC (½ inch) in comparison to HERA (¾ inch), the HALFBAC collection efficiency is quite strongly influenced by changes in wind speed and direction (see figures below).

[Figure]

The left plot shows the sampling efficiency at a constant wind speed of 1.7 m/s (same as mean wind speed during sampling periods 4 and 5) with variable aspiration angle of 0° (inlet facing wind directly) to 90° (inlet facing wind at 90°). Solid lines represent HALFBAC (½ inch inlet), dashed lines HERA (¾ inch inlet). The right plot shows the sampling efficiency at a

constant aspiration angle of 0° with variable wind speed ranging from 1 m/s to 3 m/s. In general, HERA samples particles with diameters ranging from 0 to ~8 µm, which comprises the vast majority of the urban background aerosol particle population (see measurements by Mordas et al., 2015), more efficiently than HALFBAC for the given parameters. It can be seen that an increase in aspiration angle and a decrease in wind speed cause particles to be sampled less efficiently with HALFBAC. At constant wind speed, $D_{50}$ shifts from 7.3 µm to 5 µm with a change in aspiration angle from 0° to 60° (see left plot). If the inlet is facing into the wind (aspiration angle = 0°), $D_{50}$ is shifted from 10.4 µm (3 m/s) to 5.8 µm (1 m/s). The sampling periods with the largest discrepancies in INP concentration between HERA and HALFBAC (3, 4, and 5) are also the ones with the lowest wind speed and the strongest variability in wind direction, and could thus be the periods with the least efficient sampling with HALFBAC. This information is now included in Sec. 3.3.

In summary, this is paper about an important new instrument and its characterization for INP sampling. Much good material is presented. Improvements can be gained by simplification of the paper.

References (only those not included in original manuscript reference list):

Govindarajan, A. G., & Lindow, S. E. (1988). Size of bacterial ice-nucleation sites measured in situ by radiation inactivation analysis. *Proceedings of the National Academy of Sciences*, *85*(5), 1334-1338.

Mordas, G., Prokopciuk, N., Byčenkienė, S., Andriejauskienė, J., & Ulevicius, V. (2015). Optical properties of the urban aerosol particles obtained from ground based measurements and satellite-based modelling studies. Advances in Meteorology, 2015.

Pruppacher, H. R. and Klett, J. D.: Microphysics of Clouds and Precipitation, Kluwer Academic Publishers, Dodrecht, The Netherlands., 1997.

---

## Author Comment (AC2)

**Author response to comments by Referee #2:**

We thank the referee for taking her/his time to review our manuscript and for her/his helpful comments and recommendations. Even though publication of the first manuscript version was not recommended, we hope that the referee reconsiders after reviewing our adjustments. Since HERA is currently revised and not available for further experimentation, we could not perform additional measurements as requested. However, we included more elaborate descriptions of the experimental setups, clarified particle losses in the actual CIRRUS-HL campaign setup, and estimated measurement uncertainty due to differences in aspiration efficiency when instruments were not sampling from a common inlet.

In the following, we address the referee comments and describe our adjustments in detail. For this, the referee comments are given in blue and our answers in black. When referencing page, line, and section numbers, we always refer to the first version of the manuscript, unless otherwise stated.
* * *
The manuscript describes an automated 6-filter sampling system (called HERA) for collecting atmospheric ice nucleating particles (INPs) for offline immersion freezing analysis. Results are presented for HERA at relatively modest flow rates (~40 L/min) for laboratory INPs (Arizona Test Dust, SNOMAX), ambient ground sampling, and airborne measurements on the HALO. Samples were collected alongside the HALFBAC single-filter holder, which was designed for balloon-borne observations. The experimental design for validating aspects of the HERA system in some cases does not seem particularly well posed. For example, the effort to characterize the collection efficiency across varying filter pore sizes focuses on offline filter extraction and immersion freezing INP measurements, which have their own sources of uncertainty. Quantifying size-dependent filtration efficiency of large particles could've been more straightforwardly accomplished by using a size classifier and particle counters. Similarly, the theoretical calculations of particle transmission efficiency focus on the relatively short transport tube lengths within the sampler, but neglect the important aircraft inlet and long tube lengths that are going to be the limiting factors. Overall, the manuscript is well written albeit quite long. The topic is relevant for Atmospheric Measurement Techniques. While I do see some value in having an instrument paper to describe a specific airborne instrument, I'm struggling to identify what is novel in this work or, at least, of utility to the community with regard to airborne filter sampling. Consequently, I can't recommend the paper for publication, unless substantial efforts are made to deepen the level of characterization and analysis with regard to the HERA system itself.

We agree with the referee that there are alternatives to the here presented quantification of the collection efficiency via immersion freezing measurements. We opted for this method, since HERA is first and foremost used for the collection of INPs. The comprehensive examination of INP sampling onto filters, extraction of INPs from the filter material, and immersion freezing measurements of the filter extract is the typical HERA use case. We are aware of the fact, that the immersion freezing measurements have their own uncertainties

(which are discussed in the manuscript, see p. 8, l. 195-198) and hence compared our results to published data, where good agreement was found. We included the following in Sec. 3.2 to clarify the reasoning behind our approach: "In order to verify the theoretical particle transmission efficiencies for different particle sizes and flow rates, laboratory experiments with test substances were performed. This was done via immersion INP filter analysis, which is the typical HERA use case."

Concerning the transmission efficiency calculations, we intended to strictly separate the instrument itself from the inlet and tubing, as the latter change with different campaign setups. However, we understand the request to give an example to communicate transmission efficiencies for installation of HERA on aircraft. We now include new calculations representing the CIRRUS-HL inlet and tubing setup in Sec. 4, together with the results from the aircraft filter sampling, in addition to the "HERA-only" calculations in Fig. 2.

Concerning the innovation of the new HERA system, we feel that we have described improvements compared to typically used methods in detail in the introduction (high degree of automation, no more manual filter handling, active pump control, flow rate exceeding 100 L/min and with that significantly higher spatial resolution than other systems, p. 4, l. 102-108). These technical advancements have already generated interest and request for more HERA systems from the INP community, showing the high degree of innovation. Furthermore, referee #1 judged the scientific significance as "excellent".

Specific Comments:

What aspects of the HERA make it "next generation"? Frankly, it seems like a pretty straightfoward, moderate-flowrate (40 L/min), filter sampler, albeit with multiple cartidges instead of one.

Please refer to our statement concerning the innovation of the HERA system given above. HERA can be used for filter sampling at flow rates up to 120 L/min at near-standard pressure conditions (now included in Sec. 3.1) which is way above any other filter sampling setup described in the literature (highest flow rate 50 L/min given by Flyger et al., 1973, see p. 4, l. 94-95). The flow rate of 40 L/min during CIRRUS-HL was restricted due to the amount of other instruments sampling from the same line. We added the following to Sec. 5: "Setups for past campaigns have been, and upcoming ones will continue to be, planned in such a manner that sampling flow rates are maximized and hence temporal and spatial resolution of retrieved INP concentrations further increased."

Pg. 5, Line 155: "adapt the sample flow velocity to the wind speed". This doesn't read quite right. Suggest instead something like "match the inlet face velocity to the velocity of the surrounding air (often represented as the aircraft true air speed or for stationary sampling, wind speed)"

The sentence was changed according to the suggestion.

Figure 1 is very busy and shows a lot of unrelated information. Suggest combining the HALO aircraft diagram (panel a) with Figure 2 as it adds little value about the actual HERA

instrument itself and splitting panels b and c into separate figures that are large enough to be legible. Additional labels on the filter holder cross-section, would be valuable -- what is the orange shaded region?

We considered combining Fig. 1 a) with Fig. 2 but decided against it because Fig 1 a) is meant as a schematic to introduce possible installation of HERA on aircraft, not as a to-scale sketch of sampling lines relating to transmission efficiency calculations (now included in Sec. 4). We appreciate the remark about splitting Fig. 1 b) in two and including labels for the cross section (see new Fig. 1 b and c). The old Fig. 1 c), i.e., the cold stage setups and measurement examples, were removed to concentrate on the sampling, not the filter analysis.

Following up on the previous comment, I would think that the most important sampling transmission concerns would be from the inlet(s) and long lengths of transport tubing, which Figure 2 and related discussion do not account for. This is a flaw, and Figure 2 is currently somewhat misleading in only accounting for the very short tube lengths within the instrument itself. Please include the inlet aspiration efficiency for the isokinetic and CVI inlets in the calculations as well as a rough transport tubing length. I suspect that the inflection point in Figure 2 will be shifted dramatically toward smaller sizes than what is presented now.

We understand the concern. Since Sec. 2 focuses on the instrument itself, we would like to avoid including the particle loss calculations for a specific measurement campaign there. As a compromise, we included information concerning the transmission efficiency during CIRRUS-HL in Sec. 4 in written form. The $D_{50}$ values at the HASI and HALO-CVI are 2.7 μm and 2.2 μm, respectively (see figure below). Section 4 now also contains information about the rough sampling line geometries and other parameters used for the calculations. Sec. 2.2 still focuses on the discussion of the "HERA-only" transmission efficiency (which we feel is important information, too) and refers to Sec. 4.

[Figure]

The $D_{50}$ for sampling at the HASI given above includes aspiration effects in the inlet. Concerning the HALO-CVI, however, it is not possible to include its aspiration efficiency into the tube transmission calculations. The reason is that the aspiration efficiency is related to cloud droplets and/or ice particles, whereas the tubing transmission efficiency is related to the dry cloud particle residuals, which are then sampled with HERA. These are totally different diameter regimes, which cannot be treated in the same calculation as a function of particle diameter. Similar to other CVI systems on fast flying aircraft, the design of HALO-CVI allows a lower cut-size of 5 µm (e.g., Twohy and Poellot, 2005). The upper cut size is given by the distance to stop and sublimate larger cloud particles in comparison to the inlet geometry (Czizco and Froyd, 2014). This limits the sampling without wall contact to cloud particles smaller 60 µm (Seifert et al., 2004). Thus, one can proceed from the assumption that residual particles from cloud particles in the size range of 5 µm to 60 µm leave the HALO-CVI inlet without losses from where the particle transmission calculations to HERA are carried out. While this is important information, we feel that including it in the manuscript is beyond the scope of the present study. The size range of collected cloud particles with the corresponding references is now included in Sec. 4.

Pg. 7, Line 181: How long does it take to remotely switch between the filters? Closer to a second is fast, but closer to a minute is not particularly fast for aircraft sampling.

The first version of HERA included a positioning system connected to the ball valve which would work reliably only when turned into one direction. This meant that the next filter holder in turning direction could be reached within ~10 s, but the previous filter holder in ~60 s. Now, HERA is being revised so that the positioning is accurate even when the motor is running in reverse. Furthermore the turning speed is increased. Since this new version will be in operation in the near future, we revised the time for switching to < 30 s.

Section 2.3: Can filters be swapped out in-flight in order to sample more than 6 filters?

In principle, this is possible. The filter holder inset would have to be disconnected from the sampling line in-flight, removed, and the filter holders replaced with "fresh" ones. Alternatively, a second filter holder inset (complete with filter holders, ball valves and motor drive) could be brought on board and be exchanged in-flight, which would be faster. However, this is not possible on HALO due to certification regulations. Frankly, the number of filter holders was discussed in detail prior to the construction of HERA and so far 6 filters per flight have proven to be an appropriate number in practice. It is true that one has to think very carefully about meaningful sampling intervals prior to takeoff and in-flight, but this also holds true in case of more available filters. Based on suggestions from referee #1, we have rewritten the paragraph concerning sampling strategy in Sec. 2.3 and included the information above.

Pg. 7, Line 189: The paper seems to shift at this point away from the HERA operation to discussion of offline analyses of the filter extracts. This should be a new section.

The description of the offline immersion freezing analysis was shifted to the Appendix to make it easily available to the reader but not to distract from the information about sampling.

Pg. 8, Lines 207-215: Do these reported 1-minute and 10-minute sampling time recommendations correspond to a filter flow rate of 40 L/min. Could a higher flow rate be used?

Yes, these periods refer to a sample flow rate of 40 L/min, as stated in the caption of Fig. 3 and on p. 8, l. 202. This flow rate was chosen for the calculation as it was used during CIRRUS-HL, which is a recurring example in our manuscript. Yes, a higher flow rate could be used, as the pump unit is able to generate up to 150 L/min for undisturbed standard conditions. The actual flow rate through the filter depends mostly on the filter medium and the ambient pressure (see comment on p. 7 of this document). More information on this is now included in Sec. 3.1, where the effect of filter pore size is discussed.

Pg. 10, Line 250: Now the flow rate is 15 L/min. Does this change the filter collection efficiency?

A higher flow rate will increase the collection efficiency. This is discussed on p. 11, l. 275. The flow rate of 15 L/min was chosen for practical reasons, i.e., to allow for prolonged sampling through 0.2 μm pore size filters with the battery-powered HALFBAC. A higher flow rate would have led to a significantly higher power consumption. This was added to the manuscript in Sec. 3.1.

Pg. 10, Line 255: Why 15 minutes? This strikes me as quite a long time for each filter.

This is our established method. In earlier tests, the time period of 15 min proved to remove the overall majority of particles from polycarbonate filters, except for some hydrophobic soot aggregates. This was investigated by examining a sampled filter via scanning electron microscopy prior to washing and again after washing and drying. The optimal filter washing/rinsing time most likely depends on the vessel, volume of washing water, and the

type of shaker used. In the literature, time periods of, e.g.,  3 min (Jakobsson et al., 2022), 20 min (e.g., McCluskey et al., 2017), and 1 h (e.g., Adams et al., 2020) are reported. We feel that an elaborate discussion of filter washing times is out of the focus of our study and did not change anything in the manuscript.

Pg. 10, Line 268: Why would there be differences in aspiration efficiency? Where the inlets and flow rates meaningfully different?

In this case, the inlets and flow rates were exactly the same, as described on p. 10, l., 248-250. However, variations in wind speed and direction could have affected both HALFBACs to different degrees during the rooftop sampling and influenced the overall sampling efficiency to different degrees. This was now added to Sec. 3.1. Furthermore, we now refer to Sec. 3.3, where the effect of variations in wind speed and direction on the sampling efficiency are estimated.

Pg. 14, Lines 341-342: Strike the sentence "In conclusion..." as I don't think a statement with this level of strength or clarity is appropriate to summarize what are rather messy experimental data.

In this case, we do not agree with the referee. In our opinion, the sentence in question is an appropriate summary of the sampling experiments with ATD. We do not see how the data could be interpreted as "messy" when good agreement between all sampled particle sizes at all different flow rates was found. Furthermore, our data agree with published literature data. All of this implies, that no noticeable loss of submicron ATD particles occurred and that the sampled particles were available to nucleate ice in the offline immersion freezing experiments. Nothing was changed.

Pg. 15, Lines 368-369: Strike the sentence, "From this...". The attribution of observational differences solely to differences in aspiration efficiency is speculative.

We now included additional information about the variable aspiration efficiency of HALFBAC (see comment below). According to this, the sentence was changed to: "In summary, INP concentrations and thus INP sampling efficiencies agree within measurement uncertainty for the sampling periods that presumably did not feature significant differences in sampling efficiency between HERA and HALFBAC."

Section 3: The Particle Loss Calculator can be used to compute inlet aspiration losses. Given the reported (relatively low wind speeds), what is a reasonable discrepancy associated with differences in the inlet aspiration? This could be used to uncertainties to the data in Figure 4 and Figure 7.

As suggested, aspiration and transmission efficiencies were calculated for the sampling experiments with HERA and HALFBAC (see figures below). The left plot shows the sampling efficiency at a constant wind speed of 1.7 m/s (same as mean wind speed during sampling periods 4 and 5) with variable aspiration angle of 0° (inlet facing wind directly) to 90° (inlet facing wind at 90°). Solid lines represent HALFBAC (½ inch inlet), dashed lines HERA (¾ inch inlet). The right plot shows the sampling efficiency at a constant aspiration angle of 0° with

variable wind speed ranging from 1 m/s to 3 m/s.  In general, HERA samples particles with diameters ranging from 0 to ~8 µm, which comprises the vast majority of the urban background aerosol particle population (see measurements by Mordas et al., 2015), more efficiently than HALFBAC for the given parameters. It can be seen that an increase in aspiration angle and a decrease in wind speed cause particles to be sampled less efficiently with HALFBAC. At constant wind speed, $D_{50}$ shifts from 7.3 µm to 5 µm with a change in aspiration angle from 0° to 60° (see left plot). If the inlet is facing into the wind (aspiration angle = 0°), $D_{50}$ is shifted from 10.4 µm (3 m/s) to 5.8 µm (1 m/s). The sampling periods with the largest discrepancies in INP concentration between HERA and HALFBAC (periods 3, 4, and 5) are also the ones with the lowest wind speed and the strongest variability in wind direction, and could thus be the periods with the least efficient sampling with HALFBAC. This information is now included in Sec. 3.3. It is not possible to derive implications of the described effects on the measured INP concentrations since the overall aerosol particle size distribution, let alone the size distribution of the INPs, is not known.

[Figure]

Page 16, Line 381: What was the inlet tip diameter? What was a typical total air flow rate?

The inlet tip is 8.82 mm in diameter, the total air flow varied with TAS, e.g., ~73 L/min at 200 m/s (~11 km flight altitude). This was now added is Sec. 4.

Page 16, Lines 383-384: If the CVI flow was only 5 L/min and HERA was pulling 40 L/min, where did the makeup air flow come from?

Due to its operation mode and geometry, the HALO-CVI only allows for a limited total flow which is made up of sample and supply flow. The sample flow (~ 11 L/min) is then distributed to a number of instruments, and HERA received 5 L/min. Since particles are enriched in the inlet, the lower flow rate will not strongly influence the probability to sample INPs. We added the following to Sec. 4: " The volumetric flow rate of HERA at the HALO-CVI was ~5 L/min which is due to the inlet-specific restriction of total flow rate. However, since cloud particles and hence residuals are enriched in the HALO-CVI, the lower flow rate does not decrease the probability to collect INPs in comparison to sampling at the HASI."

Page 18, Lines 416-418: Were such high flow rates investigated during this work? I would think that that would be very useful! What is the maximum flow rate for the 800-micron filters?

HERA was used for sampling SNOMAX and ATD onto 0.8 µm pore size filters at a flow rate of 100 L/min as described in detail in Sec. 3.2 (comparison of sampling efficiency at 10, 40, and 100 L/min, see Fig. 5 and 6). In these experiments, the pump unit was running at ~80 % of the maximum speed at standard pressure. Performance tests showed that just under 150 L/min would be possible at standard pressure through 0.8 µm pore size filters if the pumps are completely maxed out, which is not recommended for a prolonged time period. 120 L/min are more realistic. At low pressure (200 mbar) the pump speed increases by ~factor 2 compared to standard pressure, i.e., in this case 60 L/min can still be achieved. Note that the use of high flow rates at low pressure can lead to a disintegration of the filter material as observed for 0.2 µm pore size filters at 40 L/min and 200 mbar (see p.9, l.232-233). This issue was not observed for 0.8 µm pore size filters sampled at 40 L/min. A short version of this was now added to Sec. 3.1.

Page 18, Line 419: I am skeptical that a 7 micron particle can be efficiently sampled through an aircraft inlet and actually make it to the filter to be collected. Please provide information about particle collection efficiency that would be expected for the deployment of HERA on the HALO for CIRRUS-HL.

See adaptions in Sec. 2.2 and 4. Here we adjusted the sentence in question in the following way: "The system was designed for efficient sampling of supermicron particles at high flow rates (particle transmission in HERA: $D_{50} = 7$ µm at 40 L/min and near-standard pressure, exemplary particle transmission including aircraft inlet and sampling lines: $D_{50} = 2.7$ µm at 40 L/min and 340 mbar)."

Page 19, Line 427-428: First, how do these findings suggest efficient sampling of INPs? Second, what might cause alteration of the particles' immersion freezing properties? What has been ruled out?

1) Our results compare well with data from the literature which was retrieved from direct measurements of SNOMAX® and ATD suspensions. This means, that a similar number of INPs per mass/surface area is found in those suspensions and in the filter extracts from our sampling experiments. If INPs would be lost during sampling with HERA, this would result in a lower number of INPs per mass/surface area in comparison to the literature values. We added these explanations in Sec. 3.2, so that the statement about "efficient INP sampling" in the summary should no longer bring up any questions.

2) For example, the particles' immersion freezing properties might be altered by impaction on the filter (disintegration, making more/other surface sites available). Storage of the filters could also lead to a change in immersion freezing properties. These issues have not been ruled out directly, but indirectly by comparing to results from the literature. The sentence has been altered to include the mentioned processes.

Section 3.3, Pg. 19, Line 432-433: Since the confounding results are due to a flaw in the experimental design, could the experiment be redone so that both instruments are sampling from a common inlet?

As stated earlier, unfortunately we cannot perform additional experiments as HERA is currently being revised. By adding additional information concerning the potential effect of differences in aspiration efficiency on the collection efficiency of INPs and the results obtained during the PICNIC campaign (Lacher et al., 2023) in Sec. 3.3, we hope to have discussed potential reasons for the discrepancies in $N_{INP}$ of HERA and HALFBAC in sufficient detail.

Appendix A: I don't understand the need for this section and suggest that it be removed. Much of the discussion seems to be focused on some experiments with SNOMAX to assess the contribution of any leaks when the system is operating at the same pressure as its surroundings, and it is found that the system was not leaking. For the aircraft campaign, where the differential pressure between the cabin and the system can be significant, the manuscript merely notes that leaks are avoided by leak testing the system at low vacuum and that any transient leaks would be identified in flight.

Appendix A was removed and the part about leak testing in the laboratory shifted to Sec. 4.

References:

Adams, M. P., Tarn, M. D., Sanchez-Marroquin, A., Porter, G. C. E., O'Sullivan, D., Harrison, A. D., Cui, Z., Vergara-Temprado, J., Carotenuto, F., Holden, M. A., Daily, M. I., Whale, T. F., Sikora, S. N. F., Burke, I. T., Shim, J.-U., McQuaid, J. B., and Murray, B. J.: A Major Combustion Aerosol Event Had a Negligible Impact on the Atmospheric Ice-Nucleating Particle Population, Journal of Geophysical Research: Atmospheres, 125, e2020JD032 938, https://doi.org/10.1029/2020JD032938, 2020.

Cziczo, D. J., & Froyd, K. D. (2014). Sampling the composition of cirrus ice residuals. Atmospheric Research, 142, 15-31.

Flyger, H., Hansen, K., Megaw, W. J., and Cox, L. C.: The Background Level of the Summer Tropospheric Aerosol Over Greenland and the North Atlantic Ocean, Journal of Applied Meteorology and Climatology, 12, 161–174, https://doi.org/10.1175/1520-0450(1973)012<0161:TBLOTS>2.0.CO;2, 1973.

Jakobsson, J. K. F., Waman, D. B., Phillips, V. T. J., and Bjerring Kristensen, T.: Time dependence of heterogeneous ice nucleation by ambient aerosols: laboratory observations and a formulation for models, Atmospheric Chemistry and Physics, 22, 6717–6748, https://doi.org/10.5194/acp-22-6717-2022, 2022.

Lacher, L., Adams, M. P., Barry, K., Bertozzi, B., Bingemer, H., Boffo, C., Bras, Y., Büttner, N., Castarede, D., Cziczo, D. J., DeMott, P. J., Fösig, R., Goodell, M., Höhler, K., Hill, T. C. J., Jentzsch, C., Ladino, L. A., Levin, E. J. T., Mertes, S., Möhler, O., Moore, K. A., Murray, B. J., Nadolny, J., Pfeuffer, T., Picard, D., Ramírez-Romero, C., Ribeiro, M., Richter, S., Schrod, J.,

Sellegri, K., Stratmann, F., Swanson, B. E., Thomson, E., Wex, H., Wolf, M., and Freney, E.: The Puy de Dôme ICe Nucleation Intercomparison Campaign (PICNIC): Comparison between online and offline methods in ambient air, EGUsphere [preprint], https://doi.org/10.5194/egusphere-2023-1125, 2023.

McCluskey, C. S., Hill, T. C. J., Malfatti, F., Sultana, C. M., Lee, C., Santander, M. V., Beall, C. M., Moore, K. A., Cornwell, G. C., Collins, D. B., Prather, K. A., Jayarathne, T., Stone, E. A., Azam, F., Kreidenweis, S. M., & DeMott, P. J. (2017). A Dynamic Link between Ice Nucleating Particles Released in Nascent Sea Spray Aerosol and Oceanic Biological Activity during Two Mesocosm Experiments, Journal of the Atmospheric Sciences, 74(1), 151-166. https://doi.org/10.1175/JAS-D-16-0087.1

Mordas, G., Prokopciuk, N., Byčenkienė, S., Andriejauskienė, J., & Ulevicius, V. (2015). Optical properties of the urban aerosol particles obtained from ground based measurements and satellite-based modelling studies. *Advances in Meteorology*, *2015*.

Seifert, M., Ström, J., Krejci, R., Minikin, A., Petzold, A., Gayet, J.-F., Schlager, H., Ziereis, H., Schumann, U., and Ovarlez, J.: Aerosol-cirrus interactions: a number based phenomenon at all?, Atmos. Chem. Phys., 4, 293–305, https://doi.org/10.5194/acp-4-293-2004, 2004.

Twohy, C. H. and Poellot, M. R.: Chemical characteristics of ice residual nuclei in anvil cirrus clouds: evidence for homogeneous and heterogeneous ice formation, Atmos. Chem. Phys., 5, 2289–2297, https://doi.org/10.5194/acp-5-2289-2005, 2005.